# PertEval: Unveiling Real Knowledge Capacity of LLMs with Knowledge-Invariant Perturbations

**Jiatong Li[1][†], Renjun Hu[2], Kunzhe Huang[2], Yan Zhuang[1],**
**Qi Liu[1][*], Mengxiao Zhu[1], Xing Shi[2], Wei Lin[2]**
[1]University of Science and Technology of China, China
[2]Alibaba Cloud Computing, China
{cslijt, zykb}@mail.ustc.edu.cn, {qiliuql, mxzhu}@ustc.edu.cn,
{renjun.hrj, huangkunzhe.hkz, shubao.sx, weilin.lw}@alibaba-inc.com

## Abstract

Expert-designed close-ended benchmarks are indispensable in assessing the knowledge capacity of large language models (LLMs). Despite their widespread use, concerns have mounted regarding their reliability due to limited test scenarios and an unavoidable risk of data contamination. To rectify this, we present PertEval, a toolkit devised for in-depth probing of LLMs' knowledge capacity through **knowledge-invariant perturbations**. These perturbations employ human-like restatement techniques to generate on-the-fly test samples from static benchmarks, meticulously retaining knowledge-critical content while altering irrelevant details. Our toolkit further includes a suite of **response consistency analyses** that compare performance on raw vs. perturbed test sets to precisely assess LLMs' genuine knowledge capacity. Six representative LLMs are re-evaluated using PertEval. Results reveal significantly inflated performance of the LLMs on raw benchmarks, including an absolute 25.8% overestimation for GPT-4. Additionally, through a nuanced response pattern analysis, we discover that PertEval retains LLMs' uncertainty to specious knowledge, and reveals their potential rote memorization to correct options which leads to overestimated performance. We also find that the detailed response consistency analyses by PertEval could illuminate various weaknesses in existing LLMs' knowledge mastery and guide the development of refinement. Our findings provide insights for advancing more robust and genuinely knowledgeable LLMs. Our code is available at `https://github.com/aigc-apps/PertEval`.

## 1 Introduction

Large language models (LLMs) are developing rapidly, and have shown excellent basic capabilities, such as reasoning [1, 2], planning [3] and world knowledge [4], in real-world tasks. Given the widespread deployment of LLMs across an increasing number of scenarios, including those that are safety-critical [5], evaluating the real capability of LLMs becomes a necessary and significant task. Knowledge capacity, *i.e.,* the ability to retrieve and utilize acquired knowledge to solve professional problems, is one of the core capabilities of LLMs [4]. Existing knowledge capacity evaluation largely rely on standardized tests using close-ended benchmarks [6–11]. These benchmarks consist of multiple-choice questions that include question descriptions, goals, options, and correct answers. They encapsulate valuable domain-specific knowledge that LLMs need to comprehend. The knowledge capacity of LLMs can then be directly gauged by the performance on these test datasets.

---

[†]Work done during Li's internship at Alibaba Cloud Computing, under the guidance of Hu.
[*]Qi Liu is the corresponding author.

38th Conference on Neural Information Processing Systems (NeurIPS 2024) Track on Datasets and Benchmarks.

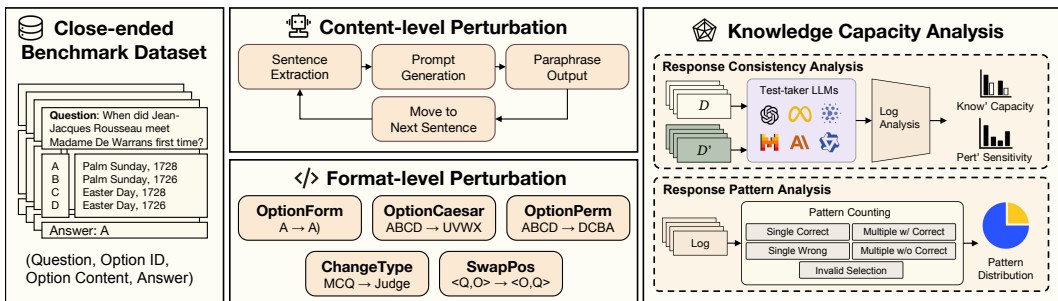

Figure 1: An overview of the PertEval evaluation toolkit. PertEval uses content-level and format-level perturbations to generate perturbed dataset $D'$ from *existing* close-ended benchmark dataset $D$. Next, it evaluates the knowledge capacity of LLMs via response consistency analysis. PertEval also demonstrates in-depth the performance feature of LLMs via response pattern analysis.

To date, static benchmarks have been essential in assessing the relative knowledge capacity of LLMs [12] because of their high quality and cost effectiveness. However, due to limited test scenarios [13] and the unavoidable risk of data contamination [14], these benchmarks still encounter significant challenges in accurately measuring the true knowledge capacity of LLMs. First, static benchmarks rely exclusively on close-ended questions in fixed formats for evaluation, which differs a lot from real-world scenarios [15]. For instance, instead of merely selecting an option, some users prefer asking LLMs to judge the correctness of options, while others may directly request the LLMs to generate an answer without providing an option list [16]. These varying prompting styles have an innegligible effect on performance [17]. Furthermore, previous work [18] has demonstrated that nonlinear or discontinuous evaluation metrics could lead to inaccurate implications about LLM capacity. Therefore, to genuinely evaluate the knowledge capacity of LLMs, it is necessary to test their performance across a variety of scenarios that mirror real-world conditions. Second, in terms of data contamination [14], knowledge capacity benchmarks usually publish their test data online to ensure transparency and reproducibility (*e.g.,* [6, 7]). However, this practice enables LLMs to memorize the test data during pre-training and alignment, which can lead to an overestimation of knowledge capacity, thus undermining the trustworthiness of evaluation results [19]. Efforts to mitigate data contamination have primarily focused on detecting such contamination [14, 20] and generating new test data [21]. Nonetheless, high-quality benchmark data contain rich knowledge that is valuable for evaluation. We suppose that it is feasible to utilize this knowledge effectively to assess the true knowledge capacity of LLMs while reducing the risk of data contamination.

To achieve this goal, we introduce PertEval, an evaluation toolkit that utilizes knowledge-invariant perturbations on static benchmarks to unveil the real knowledge capacity of LLMs, as illustrated in Figure 1. This idea stems from an analogy with human educational assessment. Like LLM evaluation, close-ended questions are prevalent in human assessments due to their versatility, cost-effectiveness and precision of measurement [22]. However, they also face challenges such as cheating [23] and a lack of variety [22]. Solutions to these issues include item personalization [24] and the creation of distractive options [25, 26]. Inspired by this, PertEval incorporates human-like **knowledge-invariant perturbations** to restate static test data to various forms and employs **response consistency analyses** to evaluate and trace the change of LLMs' performance in different test scenarios. Specifically, knowledge-invariant perturbations consist of both the content-level perturbation that extensively writes questions to minimize data contamination and a series of format-level perturbations that comprehensively cover potential real-world test scenarios. Response consistency analyses encompass basic evaluation metrics to measure the real knowledge capacity of LLMs and response pattern analysis to investigate the causes behind changes in LLM performance. Consequently, PertEval could deeply probe the LLMs' weaknesses in knowledge mastery and offer insights for their refinement.

In experiments, we re-evaluate six representative LLMs using PertEval. Capacity metric results first reveal a significant overestimation of the knowledge capacity of LLMs by static benchmarks, with an absolute 25.8% overestimation for GPT-4 and 38.8% for Gemini-1.0-Pro on MMLU [6] (results of Gemini-1.0-Pro are available at Appendix D.2). Response pattern analyses further unveil that the performance decline with PertEval is mainly caused by an increase of selecting extra incorrect choices, with an increase of up to 9.7% and 27.7% for GPT-4 and Gemini-1.0-Pro, respectively.

These findings highlight the potential for rote memorization of correct options by LLMs on static benchmarks. Detailed response consistency analyses in terms of overall performance stability and correct response consistency indicate various weaknesses in the knowledge capacity of existing LLMs, such as the vulnerability to content-level perturbation and the change of global text order.

Overall, our contributions are as follows:

- We propose an evaluation toolkit that utilizes knowledge-invariant perturbations on close-ended evaluation benchmarks to unveil the real knowledge capacity of LLMs, a significant step towards more trustworthy LLM evaluation.

- We re-evaluate the knowledge capacity of six representative LLMs using PertEval. Evaluation results not only demonstrate significantly inflated performance of LLMs by static benchmarks, but also reveal LLMs' uncertainty to specious knowledge and rote memorization to correct options.

- We demonstrate the vulnerability of various LLMs to different perturbation strategies in PertEval and provide insights for the refinement in terms of promoting LLMs' knowledge capacity.

## 2   Related Work

**Knowledge Capacity Evaluation of LLMs** Research works of the knowledge capacity evaluation LLMs consist of two lines, *i.e., evaluation benchmarks* and *evaluation methodologies*. The first line aim to design comprehensive and accurate benchmarks to quantize the knowledge capacity of LLMs. Evaluation benchmarks could be further classified into *general* or *professional* knowledge benchmarks [12]. The first category aims to evaluate the general knowledge capacity of LLMs across a wide range of domains, such as MMLU [6], C-Eval [27] and ARC [8]. The second category aims to deeply evaluate the professional capacity of LLMs in specific domains, such as MedMCQA [9] in medicine and ScienceQA [28] in science. These benchmarks depend on professional multiple-choice questions to quantitatively measure the capacity of LLMs. The other research line, *i.e.,* evaluation methodologies, aim to improve the accuracy and truthfulness of evaluation via data-driven or model-driven techniques. Along this line, recent studies have emphasized data-contamination detection techniques [14, 20] and contamination-free test data generation [21, 29]. In addition, psychometric-based techniques [30], dynamic evaluation techniques [31, 32], and perturbation-based evaluation techniques such as CheckList [33] and PolyJuice [34] have been proposed to comprehensively evaluate the capability of language models (LMs) from different aspects. However, these methodologies are unsuitable for knowledge capacity evaluation, and are hard to utilize valuable information in expert-designed datasets for trustworthy evaluation.

**Consistency of LMs** The consistency of an LM denotes its behavior invariance under meaning-preserving alternations in its input [35], which is significant in natural language understanding. Since the emergence of pretrained language models (PLMs), massive efforts have been made to evaluate the consistency of LMs on various fields. Elazar et al. [35] revealed the poor consistency performance of PLMs on factual knowledge. Fierro and Søgaard [36] extended the study to multilingual PLMs and obtained similar findings. Jang et al. [37] proposed a taxonomy for various concepts of consistency and established a benchmark, BECEL, to evaluate the consistency of PLMs. As LLMs develop swiftly, Wang et al. [38] proposed to use self-consistency decoding to empower the Chain-of-Thought reasoning ability of LLMs. Recently, Rajan et al. [39] proposed KonTest, an autonomous evaluation framework that utilizes knowledge graphs to generate test samples, to probe the inconsistency in LLMs' knowledge of the world. In summary, these research works view consistency as a research subject that needs to be measured or intervened. Differently, in our study, the consistency of LLMs is viewed as a technical method, *i.e.,* a measurement for probing LLMs' real knowledge capacity.

**Adversarial Text Attacks on LMs** These techniques aim to mislead language models to yield wrong or toxic outputs with small text perturbations, which play an indispensable role in the research of robustness and safety of language models. Such efforts include but not limited to jailbreaking [40] and text classification attack [41]. Text attacks could also be classified into character-level [42, 43], word-level [44] and sentence-level [45]. Recently, LLM-based perturbations like PromptAttack [41] have also been introduced. However, considering their cost and the comprehensiveness, existing text attack methods are insufficient for the purpose of this work. Moreover, our PertEval is built upon a new concept of knowledge-invariant perturbation, which is an important supplement to this topic.

## 3 Methodology

### 3.1 Knowledge-invariant Perturbations

We first present the two types of knowledge-invariant perturbations to restate static test questions. The knowledge-invariant property of these perturbations will be discussed in the next subsection.

**Content-level perturbation: knowledge-invariant paraphrasing**. The goal of content-level perturbation is to substantially alter the phrasing of questions while retaining the original knowledge, thereby mitigating data contamination in the test data. The key challenge of such knowledge-invariant paraphrasing is to preserve the original knowledge while changing the statement as much as possible. A test question is a composite of sentences, with each provide either backgrounds, conditions or goals of the question. Therefore, to preserve knowledge of the original question, we propose a sentence-by-sentence paraphrasing algorithm using an LLM rewriter. Formally, let $q = (s_1, s_2, \ldots, s_T)$ represent the question text, where $s_t$ denotes the $t$-th sentence (for $t = 1, 2, \ldots, T$). The semantic of $s_t$ (for $t \geq 2$) depends on its *prerequisite sequence* $(s_1, \ldots, s_{t-1})$. To rewrite the entire question, an LLM rewriter is instructed to paraphrase sentence by sentence given each sentence and its original prerequisite sequences. Detailed descriptions of the paraphrasing algorithm, the prompt template for the rewriter LLM, and an example of the perturbation can be found in Appendix B.1.

**Format-level perturbation: question format refactoring**. The objective of question format refactoring is to assess the robustness of LLMs' knowledge capacity under complicated test conditions. To achieve this, we have developed a variety of format-level perturbation strategies to comprehensively evaluate the resilience of LLMs' knowledge capacity in various test scenarios.

- *Option permutation (OptionPerm).* OptionPerm reorders the contents of options while maintaining the original order of option IDs. Its goal is to evaluate *option ordering bias* in the knowledge acquisition of LLMs. By default, OptionPerm reverses the order of option contents to completely disrupt their local sequence.

- *Option format refactoring (OptionForm).* OptionForm modifies the format of option IDs, such as by appending a right parenthesis to the end. This perturbation aims to assess the *option format bias* in the knowledge acquisition of LLMs. OptionForm may influence LLM performance by altering the dependency between different tokens within the options.

- *Option ID shifting (OptionCaesar).* OptionCaesar shifts the ASCII value of option IDs to change their character, which is similar to Caesar encryption. This technique aims to investigate the *selection bias* in the knowledge acquirement of LLMs, a phenomenon empirically observed in some models. By replacing common option IDs with less common ones (*e.g.,* A/B/C/D → U/V/W/X), OptionCaesar allows us to observe whether the values of the IDs impact the LLMs' performance.

- *Question type changing (ChangeType).* ChangeType converts a multi-choice question into a multi-judgment question. This perturbation aims to examine the *question type bias* in LLMs. Since the feasible solution space of a multi-choice question and the corresponding multi-judgement question is identical (given $N$ options, the size of the feasible solution space is $2^N - 1$), an LLM that robustly acquire knowledge/skills in a question should be insensitive to ChangeType.

- *Question position swapping (SwapPos).* SwapPos switches the position of the question text and the options. It aims to evaluate the *global ordering bias* of LLMs. For rational human test-takers, SwapPos does not affect performance, as it does not alter the question content. However, SwapPos can significantly change the output distribution of self-regressive text generation models by disrupting the global ordering of input prompts.

Examples of format-level perturbations are available at Appendix B.2.

### 3.2 Knowledge Invariance Verification

We next investigate the knowledge-invariance property of the proposed perturbations.

**Knowledge invariance scoring**. This approach checks the knowledge invariance of perturbations from the perspective of humans' and LLMs' perception. Specifically, based on previous works in the measurement of semantic similarity [46] and characteristics of close-ended benchmarks, we first propose standards of knowledge-invariance scoring, as shown in Table 1. Next, we recruit

Table 1: Standards of knowledge-invariance scoring.

| Standard Name | Standard Description |
|---|---|
| Semantic Information Invariance | The perturbed question must have the same semantic information as the original question, which cannot change the name of entities, logic of statements and meaning of equations. |
| Reasoning Invariance | A human test-taker's reasoning process to obtain his/her response in the perturbed question should be consistent with that in the original question. |
| Answer Invariance | The answer of a perturbed question should be semantically equivalent to the answer of the original question. |
| Statement Clarity | The perturbed question should clearly present contexts, conditions and the target of the question. |

professional human volunteers and utilize superior LLMs, such as claude-3-sonnet, respectively, to serve as the referee to rate knowledge invariance scores. Given a set of original and perturbed question pairs $D_{dual} = \{(q_i, q_i') \mid i = 1, 2, \ldots, |D|\}$, we construct scoring prompts based on a predefined template, knowledge invariant standards, and the scoring criteria for knowledge invariance judgement (see Table 9 in Appendix B.1). After collecting the output scores from the referee LLM using these scoring prompts, we calculate the average score as the final result.

**Testing on mastered questions for LLMs**. This approach evaluates knowledge invariance based on the output performance of LLMs. The rationale here is that if a perturbation is knowledge-invariant, an LLM's performance on questions that it has truly mastered should remain consistent between the original and perturbed versions. Identifying *mastered questions* for LLMs, however, poses a challenge. To this end, we propose using questions that most LLMs can correctly answer as a proxy for mastered questions. An LLM, such as gpt-4-turbo, is then required to answer both the original and perturbed versions of these questions to validate the knowledge invariance of the perturbation. In summary, the test-based knowledge invariance checking procedure involves the following steps:

1. Given a set of LLM test-takers and the evaluation dataset $D$, construct the *mastered question set* $D_{simple} \subset D$ consisting of questions that all LLMs could answer correctly.

2. Apply a perturbation to each sample in $D_{simple}$ to create the perturbed dataset $D'_{simple}$.

3. Assess the performance of a knowledgeable LLM on both $D_{simple}$ and $D'_{simple}$. If the test results show no significant difference in performance, the perturbation can be considered knowledge-invariant.

This method emphasizes the impact of perturbations on the question-answer process of LLMs. In comparison to the scoring test, this approach is more appropriate for evaluating the knowledge invariance of content-level perturbations. We present detailed results of both tests in Experiments.

### 3.3 Response Consistency Analyses for Measuring Real Knowledge Capacity

We further devise a suite of response consistency analyses that, by comparing performance on raw vs. perturbed test sets, unveils the real knowledge capacity of LLMs. These include a metric for quantifying calibrated knowledge capacity and the more fine-grained ones helpful for revealing the vulnerability of LLMs to different perturbation strategies.

**Metric of real knowledge capacity.** To measure real knowledge capacity of LLMs, we propose the *Consistent Accuracy (ACC@Consist)* as the evaluation metric. The rationale is that if an LLM has truly mastered a question and its underlying knowledge, its performance should remain consistent across all versions of the question, including both the original and the most complex perturbed versions. Formally, let $M(\cdot)$ denotes the response function of an LLM. Let $x = (q_x, y_x) \in X$ represent a test question, where $q_x$ and $y_x$ denote the question text and the correct answer(s) of $x$. Further let $\sigma^* : X \rightarrow X$ be the most complicated composite knowledge-invariant perturbation and

Table 2: **Knowledge invariance scores↑ rated by human scorers**. Four independent scores from different human scorer groups are presented in ascending order for each cell.

| Method | C-Math | W-History | P-Psychology | P-Medicine |
|---|---|---|---|---|
| PromptAttack | 2.3/2.4/3.0/3.9 | 2.2/2.2/2.3/2.8 | 1.3/2.4/2.8/2.8 | 1.6/2.5/3.5/3.6 |
| **PertEval (ours)** | **3.6/3.8/3.9/3.9** | **3.7/4.1/4.1/4.3** | **4.3/4.4/4.5/4.7** | **4.2/4.3/4.4/4.6** |

Table 3: **Knowledge invariance scores↑ rated by superior LLMs**. Values (a/b/c) in each cell denotes the average knowledge invariance score rated by gpt-4-turbo, claude-3.5-sonnet, and llama-3.1-405b, respectively.

| Method | C-Math | W-History | P-Psychology | P-Medicine |
|---|---|---|---|---|
| PromptAttack | 3.2/3.6/3.6 | 3.2/3.3/3.7 | 3.9/3.9/3.7 | 4.1/4.3/4.2 |
| **PertEval (ours)** | **3.8/3.9/4.0** | **4.0/4.2/4.0** | **4.0/4.4/4.0** | **4.1/4.4/4.0** |

$D = \{x_1, x_2, \ldots, x_{|D|}\}$ the raw test set. Then ACC@Consist is defined as:

$$\text{ACC@Consist}(M, D) = \frac{1}{|D|} \sum_{x \in D} I[M(q_x) = y_x \wedge M\left(q_{\sigma^*(x)}\right) = y_{\sigma^*(x)}]. \tag{1}$$

Here $I(\cdot)$ is the binary indicator function. Taking a step further, we implement a response pattern analysis that uncovers the causes of the discrepancy between ACC@Consist and the original ACC.

**Overall performance stability**. In terms of performance stability, we expect that LLMs which have robustly acquired the knowledge and skills required by benchmarks should exhibit stable performance when faced with knowledge-invariant perturbations. To evaluate this, we propose *Performance Drop Rate (PDR)* as a metric to measure the overall performance stability of LLMs under knowledge-invariant perturbations.:

$$\text{PDR}(M, D, \sigma) = \frac{1}{|D|} \sum_{x \in D} \left( I[M(q_{\sigma(x)}) = y_{\sigma(x)}] - I[M(q_x) = y_x] \right). \tag{2}$$

Here $D$ denotes the original test dataset and $\sigma$ is a perturbation strategy. Essentially, $PDR$ measures the discrepancy between the LLM's accuracy on $D$ and the perturbed dataset. When $PDR < 0$, the perturbation decreases the overall performance of an LLM, indicating that it does not robustly acquire knowledge and skills. To obtain more reliable conclusions, we further conduct *Wilcoxon signed-rank test* for original and perturbed question sample pairs.

**Correct response consistency**. We also adopt the *Recall of Performance (ROP)* metric to measure correct response consistency. The term "ROP" draws an analogy to the classical recall score. Let $CC$ denote the number of responses consistently correct before and after perturbation, and $IC$ the number of responses initially correct and later incorrect after perturbation (see Appendix B.3 for detailed demonstration). Similar to the recall score, ROP is defined as the ratio of consistent correct responses to the total number of correct responses before perturbation, *i.e.,*

$$\text{ROP}(M, D, \sigma) = CC/(CC + IC) \tag{3}$$

The value of ROP is within $[0, 1]$, with higher ROP indicating better correct response consistency under perturbation $\sigma$. It should be noticed that, unlike PDR, there does not exist an absolute threshold for ROP to evaluate goodness. Rather, ROP serves as an index of correct response consistency, which can be utilized to compare the performance of different LLMs under perturbations.

## 4 Experiments

**Datasets.** We choose the test data in *College Mathematics (C-Math), World History (W-History), Professional Psychology (P-Psychology) and Professional Medicine (P-Medicine)* from the Massive Multitask Language Understanding (MMLU) [6] for evaluation. This is a trade-off between comprehensively covering evaluation domains/subjects and keeping the evaluation cost affordable. Note

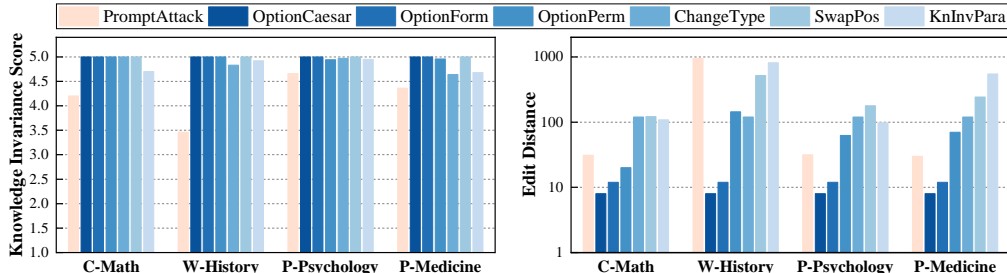

Figure 2: (left) **Perturbation-wise knowledge invariance scores↑ by gpt-4-turbo** (systematic sampling, interval = 10); (right) **edit distances** between original question and perturbed questions.

Table 4: **Results of testing on mastered questions - Performance Drop Rate (PDR)** of **overall performance stability testing on mastered questions** using gpt-4-turbo.

| Strategy | C-Math | W-History | P-Psychology | P-Medicine | AVG$_{macro}$ |
|---|---|---|---|---|---|
| KnInvPara | 0.0000 | -0.0115 | 0.0091 | -0.0244 | -0.0067 |
| OptionPerm | 0.0000 | 0.0000 | 0.0000 | -0.0244 | -0.0061 |
| OptionForm | 0.0000 | 0.0000 | 0.0000 | 0.0000 | 0.0000 |
| OptionCaesar | 0.0000 | 0.0000 | 0.0091 | 0.0000 | 0.0023 |
| ChangeType | 0.0000 | 0.0000 | 0.0000 | 0.0000 | 0.0000 |
| SwapPos | 0.0000 | -0.1149 | -0.0636 | -0.0488 | -0.0568 |

that each dataset represents a supercategory in STEM, Humanities, Social Sciences, and Other fields, respectively. The statistics of the selected datasets are detailed in Table 11 in Appendix C.1.

**Large Language Models.** Based on leaderboards like OpenCompass [47] and considering both the popularity and timeliness of LLMs, we select six representative LLMs for evaluation. These LLMs include close-sourced (gpt-4-turbo [48], gpt-3.5-turbo [49], gemini-1.0-pro [50] and glm-3-turbo [51]) and open-sourced (mistral-7b-instruct-v0.2 [52] and llama-3-8b-instruct [53]) models.

### 4.1 Knowledge Invariance Verification for Perturbations

We first investigate the knowledge invariance property of our perturbations using the two methods developed in Section 3.2. Employing the LLM-based knowledge invariance scoring method, we compare the scores of the proposed perturbations with those by PromptAttack [41] as a baseline.

**Experiment Setup**. For knowledge invariance scoring, 10 samples at equal intervals for each subject are used for scoring. Since there are four subjects and two perturbation strategies (1 baseline + 1 PertEval) for each question, we have 10*4*2 = 80 question pairs in total. To ensure that each question pair have four independent scores, eight human scorers are engaged in human-based scoring. Human scorers are equally divided into four groups. Each group independently scores for all 80 question pairs. Therefore, each human scorer within a group scores for 40 shuffled samples. For LLM-based scoring, we choose gpt-4-turbo, claude-3.5-sonnet, and llama-3.1-405b as scorers. For fine-grained perturbation-wise scoring, we choose gpt-4-turbo as the scorer.

**Result Analysis**. Overall knowledge invariance scores are presented in Table 2 and 3. PertEval outperforms the baseline, and the score mostly exceeds 4.0, the borderline of knowledge-invariance. Scores of both PromptAttack and PertEval on C-Math do not exceed 4.0. This is because many samples in College Mathematics of MMLU are short mathematical reasoning questions that have many mathematical symbols and statements. Indeed, in many STEM subjects, requirements for knowledge capacity and reasoning ability often mix together in a test question. This points out considerable potential for future development in robust LLM evaluation in STEM subjects.

Next, the results presented in Figure 2 show that the average scores of our perturbations approach 5.0 (perfectly knowledge-invariant) in most cases, consistently outperforming those from PromptAttack. We also present the Levenshtein distance (*i.e.,* edit distance) between the perturbed and original

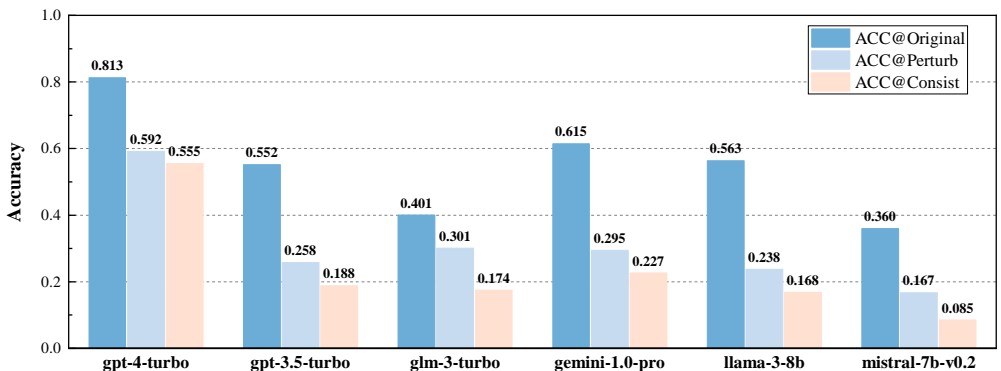

Figure 3: **Real knowledge capacities measured by ACC@Consist↑ with composite knowledge-invariant perturbation**. ACC@Original and ACC@Perturb denote accuracy on the original and perturbed data, respectively. We report the macro-averaged results on all tested datasets.

questions under each perturbation. It turns out that the traditional edit distance is not suitable for measuring knowledge invariance due to its lack of correlation with knowledge invariance.

Within mastered question testing (results depicted in Table 4), we observe that most perturbations achieve an average PDR closed to zero, indicating consistent performance on the perturbed and the original data. The PDR of SwapPos on World History and Professional Psychology is less than -0.05 despite that it does not alter any knowledge-relevant information by design. A possible explanation is that SwapPos changes the global ordering of question prompts, thus affecting the token generation of these self-regressive LLMs. More detailed analyses of results could be found in Appendix D.

## 4.2 Real Knowledge Capacity Evaluation

Each knowledge-invariant perturbation in PertEval targets a specific question restatement strategy. To construct the most challenging test scenario and obtain reliable and comprehensive evaluation results, we compose all perturbations to create a *composite perturbation*. The real knowledge capacity of LLMs is then quantified by ACC@Consist before and after the composite perturbation. The results, shown in Figure 3, yield several insights regarding the evaluation of LLMs' knowledge capacity.

**1. Overvaluation on static benchmarks**. The knowledge capacity of LLMs is significantly overvalued on static benchmarks. By comparing ACC@Original with ACC@Perturb and ACC@Consist for each LLM, we observe a notable performance drop when using the perturbed dataset. For instance, the performance of gpt-4-turbo, gpt-3.5-turbo, and gemini-1.0-pro decreases by more than 25%. Even the most capable LLM in our selection, gpt-4-turbo, achieves an ACC@Consist of only 0.555. These findings reveal a substantial gap between the evaluated knowledge capacity of LLMs on static datasets and their real knowledge capacity in complex scenarios.

**2. Robustly mastered knowledge**. Each LLM does have robustly mastered certain knowledge. This conclusion is drawn by comparing the LLMs' performance to pure guessing. With $k$ options and a single correct answer, the expected ACC@Consist for random guessing is $1/k^2$, which equals to 0.0625 for $k = 4$ in our experiments (see Appendix D.1 for detailed demonstration). We observe from Figure 3 that the ACC@Consist values of all tested LLMs are significantly higher than 0.0625, indicating that each LLM has indeed mastered some knowledge consistently. However, most LLMs have mastered less than half of the total knowledge, as indicated by ACC@Consist values below 0.5.

## 4.3 Response Pattern Analysis

We further conduct a response pattern analysis to examine how perturbations affect the performance of LLMs. We find that knowledge-invariant perturbations affect LLMs by increasing the ratio of selecting extra incorrect options. As shown in Figure 4, the main reason for the significant performance drop in gpt-4-turbo is their increased frequency of selecting additional incorrect options on the perturbed data (*i.e.,* $12\% \rightarrow 21.7\%$). These LLMs tend to select extra incorrect options alongside the correct ones. One possible explanation for this phenomenon is that **the LLMs indeed**

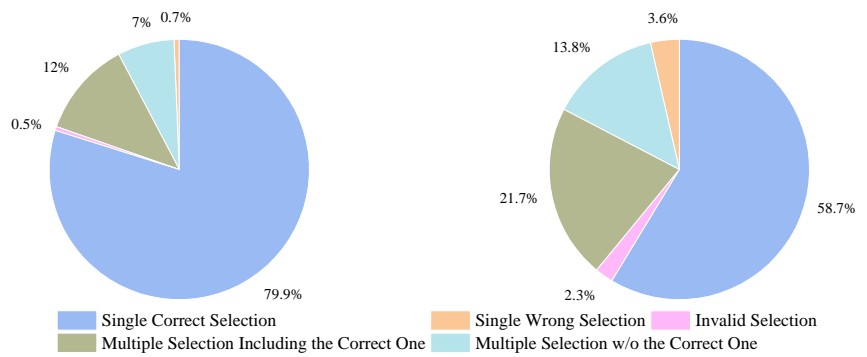

Figure 4: Response patterns of **gpt-4-turbo**. Left: original data; Right: perturbed data.

Table 5: **Macro PDR** ↑ and **hypothesis test results of Micro PDR** of LLMs w.r.t. perturbation.

| Model\Strategy | KnInvPara | OptionPerm | OptionForm | OptionCaesar | ChangeType | SwapPos | AVG |
|---|---|---|---|---|---|---|---|
| **gpt-4-turbo** | -0.0660** | -0.0208** | -0.0136 | -0.0294** | -0.0210 | -0.1117** | -0.0438 |
| **gpt-3.5-turbo** | -0.0275** | -0.0042 | -0.1767** | -0.0396** | -0.1736** | -0.1943** | -0.1027 |
| **gemini-1.0-pro** | -0.0558** | +0.0121 | +0.0125 | +0.0030 | -0.1310** | -0.1532** | -0.0521 |
| **glm-3-turbo** | -0.0370** | -0.0190 | -0.1397** | -0.0118 | +0.0522 | -0.2142** | -0.0616 |
| **mistral-7b-v0.2** | -0.0264 | -0.0200 | -0.2789** | +0.0793 | -0.0844** | -0.1275** | -0.0763 |
| **llama-3-8b** | -0.0336** | -0.0091 | -0.0939** | -0.0368** | -0.2920** | -0.1814** | -0.1078 |
| **AVG** | -0.0411 | -0.0102 | -0.1151 | -0.0059 | -0.1083 | -0.1637 | |

$^{**}$: The *Micro* PDR is significantly negative in the Wilcoxon signed-rank test ($\alpha = 0.01$).

**lack the ability to distinguish uncertain knowledge and filter out distracting options, while they correctly answer these questions in the original dataset via rote memorization. In other words, these LLMs do not truly master the knowledge behind the questions**. Therefore, it is crucial to not only enhance LLMs' ability to identify correct answers but also bolster their capability to recognize and eliminate incorrect answers when confronted with distracting choices in real-world scenarios. More detailed response pattern analysis results can be found in Appendix D.2.

### 4.4 Overall Performance Stability

Table 5 presents the overall performance stability results, in which each number is the *macro*-PDR, *i.e.,* the average of PDRs separately calculated on each dataset. We also apply the Wicoxon signed-rank test to the *micro*-PDR, which is the PDR calculated across all datasets combined. Specifically, let $s = I(M(q_x) = y_x) \in \{0, 1\}$ and $s' = I(M(q_{\sigma(x)}) = y_{\sigma(x)}) \in \{0, 1\}$ denote model $M$'s score on the original and perturbed questions, respectively. The alternative hypothesis of the Wilcoxon signed-rank test is $H_a : s > s'$, or equivalently, $H_a : s' - s < 0$. We can analyze the results of Table 5 from both a column-wise and row-wise perspective. From the column-wise perspective, we first observe that all selected LLMs have negative macro-PDRs given the content-level perturbation, *i.e.,* knowledge-invariant paraphrasing (KnInvPara). Additionally, the micro-PDR of close-sourced LLMs is *significantly* negative, meaning that these LLMs lack robustness in content-level knowledge acquisition for these datasets. Another prominent observation is that all LLMs are highly sensitive to the SwapPos perturbation. We hypothesize that this sensitivity is due to the global format change introduced by SwapPos, which disrupts the text generation process of self-regressive LLMs, even though the perturbation is entirely knowledge-invariant. From the row-wise perspective, we find that our perturbations cause performance drop in most cases, *e.g.,* at least 0.04 and up to 0.1 on average, indicating the universal vulnerability of current generation LLMs.

### 4.5 Correct Response Consistency

The results of correct response consistency using ROP are presented in Table 6. There is significant variation in ROP across different LLMs. Among them, gpt-4-turbo performs the best, with an average macro-ROP of 0.9, which is a substantial advantage over other LLMs. Gemini-1.0-pro also performs well, particularly with the OptionForm and OptionCaesar strategies (both having a Macro-ROP >

Table 6: **Macro Recall of Performance (ROP)** ↑ of LLMs w.r.t. perturbation strategies.

| Model\Strategy | KnInvPara | OptionPerm | OptionForm | OptionCaesar | ChangeType | SwapPos | AVG |
|---|---|---|---|---|---|---|---|
| **gpt-4-turbo** | 0.8798 | 0.9063 | 0.9601 | 0.9349 | 0.9221 | 0.7995 | 0.9005 |
| **gpt-3.5-turbo** | 0.8230 | 0.7728 | 0.5602 | 0.7801 | 0.5194 | 0.5610 | 0.6694 |
| **gemini-1.0-pro** | 0.7968 | 0.7995 | 0.9553 | 0.9226 | 0.6788 | 0.6362 | 0.7982 |
| **glm-3-turbo** | 0.7164 | 0.6756 | 0.5943 | 0.7884 | 0.6816 | 0.4026 | 0.6432 |
| **mistral-7b-v0.2** | 0.6543 | 0.6202 | 0.1339 | 0.7679 | 0.5128 | 0.4060 | 0.5159 |
| **llama-3-8b** | 0.8055 | 0.7433 | 0.7541 | 0.8102 | 0.3971 | 0.5209 | 0.6719 |
| **AVG** | 0.7793 | 0.7530 | 0.6597 | 0.8340 | 0.6186 | 0.5544 | |

0.9). Other LLMs, however, are vulnerable to almost all knowledge-invariant perturbations, which means that their performance on the original datasets are less reliable. On the other hand, the effect of different strategies on correct response consistency varies considerably. Similar to the results for performance stability, SwapPos is the most influential strategy affecting LLMs' correct response consistency, which exposes the inherent flaw of LLMs in defending global order perturbation.

## 5  Discussion

**Conclusion.** The trustworthiness of capacity evaluation is, and will always be one of the most significant issues in the development of LLMs. In pursuit of this goal, we proposed an evaluation toolkit, PertEval, for unveiling LLMs' real knowledge capacity on close-ended benchmarks. We discovered not only a substantial overestimation of the knowledge capacity of LLMs by static benchmarks, but also LLMs' uncertainty to distinguish specious knowledge without rote memorization. Detailed response consistency analyses revealed the vulnerability of LLMs to various knowledge-invariant perturbations, especially the content-level KnInvPara and the format-level SwapPos, which could inspire further refinement for LLMs' knowledge mastery. Indeed, *the ultimate goal of this study is not only trustworthy evaluation, but to propel the development of LLMs' knowledge capacity.* Our primary experiments have verified the feasibility of using PertEval-generated data to improve LLM robustness to various test conditions via supervised fine-tuning (see Appendix D.3). We believe that PertEval marks a significant step toward more trustworthy LLM evaluation.

**Limitations & Future Work.** The first limitation of this work is its scope regarding suitable benchmarks. PertEval is suitable for close-ended benchmarks and cannot be directly applied to open-ended ones. The challenges stem from the variety of open-ended questions and the difficulty of their objective evaluation. Future research could explore how the concept of knowledge-invariant perturbation can be adapted to improve the trustworthiness of open-ended evaluations. The second limitation of PertEval is the lack of controllability, a common issue in LLM-based text generation. Due to the complexity of LLMs, it is challenging to precisely control specific characteristics in paraphrased texts while keeping knowledge-invariance. In the future, we aim to achieve more fine-grained controllability in PertEval, such as quantitatively adjusting paraphrasing tone and length, to obtain more continuous and nuanced evaluation results of LLMs. Besides, all conclusions of this study are based on experiments conducted on a subset of MMLU due to budget constraints. We are seeking economic methods to expand our experiments and further substantiate our findings. Finally, the risk of adaptive optimization against PertEval exists as it is an open-sourced toolkit. However, the diversity of perturbations and the cost of adaptive optimization could effectively reduce the influence of adaptive optimization. By constructing a PertEval open evaluation platform, we can regularly develop and change perturbation strategies to prevent this procedure.

## Acknowledgements

This research was supported by grants from the Joint Research Project of the Science and Technology Innovation Community in Yangtze River Delta (No. 2023CSJZN0200), the National Natural Science Foundation of China (No. 62337001), the Fundamental Research Funds for the Central Universities, and the Alibaba Research Intern Program. We thank all the volunteers for their massive efforts in supporting our experiments, including Yi Cheng, Xiaowen Zhang, and Anyu Chen at Alibaba Cloud Computing, Dingchu Zhang at Nanjing University, Yuyang Xu at Zhejiang University, and Yangyang Wang at Shandong University.

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

# Appendix

# A Related Work

This section complements related works especially in LLM evaluation for Section 2.

## A.1 Benchmarks for Knowledge Capacity Evaluation

LLMs' knowledge capacity evaluation plays a fundamental yet significant role in LLM evaluation tasks. The knowledge capacity evaluation aims to quantify LLMs' ability to master and utilize professional knowledge and skills to solve real-world problems. Therefore, existing benchmarks usually depend on expert-designed test datasets, especially multiple choice questions, to evaluate LLMs' knowledge capacity.

In terms of general knowledge benchmarks, MMLU [6] proposed to comprehensively evaluate LLMs' professional knowledge on 57 subjects using the classical multiple choice question-based test. Similarly, C-Eval [27] focuses on evaluating LLMs' advanced knowledge and reasoning abilities in Chinese context. AGIEval [7] collects human-centric datasets from real-world examinations such as SAT, GRE and GMAT to evaluate LLMs' knowledge capacity in human-centric tests. Differently, ARC [8] measures LLMs' knowledge capacity from knowledge styles rather than knowledge domains, including but not limited to "Definition", "Basic Facts & Properties", "Algebraic", etc. In a word, general knowledge benchmarks usually consist of a knowledge taxonomy and a series of corresponding labelled datasets, each of which focusing on evaluating a specific aspect or domain of knowledge capacity for LLMs.

In terms of specific and professional knowledge benchmarks, expert-designed datasets are also the top priority. For clinical knowledge, MedMCQA [9], and PubMedQA [54] utilize professional multiple choice questions to assess LLMs' clinical knowledge capacity and disease diagnosis ability. For social science, Ziems et al. [55] explored evaluating LLMs' knowledge capacity in computational social science using real-world classification tasks. Nay et al. [10] explored LLMs' knowledge capacity in tax law with a case study using automatically generated multiple choice questions. For science and engineering subjects, Arora et al. [56] proposed JEEBench, a challenging scientific problem solving benchmark consisting of mathematics, physics and chemistry problems. Other representative benchmarks such as GSM-8K [11] and ScienceQA [28], also utilize expert-designed professional test data to assess LLMs knowledge capacity from various perspectives.

## A.2 Trustworthiness of Knowledge Capacity Evaluation

Large language models are self-regressive text generation models trained on large-scale online-accessible textual data. Therefore, from both the test data acquisition perspective and the response generation perspective, there exist various risks in the trustworthiness of LLMs' knowledge capacity evaluation. These risks highly threaten the correctness and reliability of the knowledge capacity evaluation results.

From the test data acquisition perspective, the test data contamination problem has been proved widely exist in evaluation benchmarks [57–60]. Test data contamination can lead to overvalued knowledge capacity, and should be avoided at all costs. To this end, much effort have been devoted into distinguishing contaminated data or preventing data contamination with new test data. Magar and Schwartz [61] studied the problem for BERT models, and proposed a principled method to analyze the exploitation and memorization of contaminated data. Jacovi et al. [60] suggested using pre-evaluation intervention such as encrypting test data and avoiding uploading textual data to the internet to prevent data contamination. On the other hand, Bai et al. [29] proposed to utilize an examiner LLM to generate test questions and evaluate LLMs' knowledge capacity to avoid data contamination. Recently, Golchin and Surdeanu [20] and Oren et al. [14] respectively proposed multi-level and example order-inspired methods for identifying test data contamination. In a word, existing works usually prevent data contamination by avoiding testing on contaminated data or generating brand-new test data. However, how to avoid data contamination while sufficiently utilize valuable knowledge information of existing test data for evaluation is still underexplored.

From the response generation perspective, LLMs' biases towards specific prompt features in knowledge capacity evaluation have been widely observed [62–67] which can highly influence their

performance. Pezeshkpour and Hruschka [62] discovered that LLMs' output sensitivity to specific ordering of options. For instance, some LLMs prefer selecting the first option in the option list, which proved the ordering bias in multiple choice question-based evaluation. Li et al. [64] further proposed methods to quantify consistence and confidence of LLMs' accuracy on test datasets. Khatun and Brown [67] explored the usefulness of multiple choice question-based dataset for evaluating open-source LLMs, and found that many small-scale open-source LLMs fail to properly understand and select an answer from given choices. Recently, Zheng et al. [63] detected the selection bias towards options in multiple choice question answering for LLMs, and proposed PriDe, a label-free and inference time-free debiasing method. However, as an external plug-in for LLMs, this method is inappropriate for application to measuring the knowledge capacity of large language models themselves. In summary, existing research usually emphasize empirically detecting biases in the response generation procedure and proposing debiasing methods. Howver, how to avoid such bias to evaluate LLMs' real knowledge capacity remains a research question.

# B   Details of Methodology

## B.1   Content-level Perturbation: Knowledge-invariant Paraphrasing

The algorithm of knowledge-invariant paraphrasing using LLM rewriter is presented in Algorithm 1. The prompt template for the LLM rewriter is presented in Figure 5. In implementation, an expected similarity score is provided to the LLM rewriter to control the degree of the change of question text. In experiments, the expected similarity score is fixed to 0.6. An example of knowledge-invariant paraphrasing in MMLU college mathematics is shown in Table 7.

---

**Algorithm 1** Knowledge-invariant Paraphrasing Using LLM Rewriter

---

**Input**: $q = (s_1, s_2, \ldots, s_T)$, the question text; $M$, the LLM rewriter
**Output**: $q' = (s'_1, s'_2, \ldots, s'_T)$, the perturbation output
1: **procedure** KNOWLEDGEINVARIANTPARAPHRASE$(q, M)$
2:     Initialize rewriting prompt template $p_{template}(\cdot)$
3:     $q' \leftarrow \emptyset$
4:     **for** $t \in 1, 2, \ldots, T$ **do**
5:         $p_{current} \leftarrow p_{template}(s_t, (s_1, \ldots, s_{t-1}))$         ▷ Generate the rewriting prompt
6:         $r_{current} \leftarrow M.request(p_{current})$         ▷ Obtain the LLM rewriter's output
7:         $s'_t \leftarrow sentenceFilter(r_{current})$
8:         $q' \leftarrow q' \oplus s'_t$         ▷ Append the perturbed sentence to the end of the question
9:     **end for**
10:     **return** $q'$
11: **end procedure**

---

Table 7: An example of knowledge-invariant paraphrasing of a test question. Texts surrounded by angular brackets are invisible in question prompts input to the LLM test-taker.

| Original | Knowledge-invariant Paraphrasing |
|---|---|
| <# **Context & Condition**> Let $T : R^2 \to R^2$ be the linear transformation that maps the point (1, 2) to (2, 3) and the point (-1, 2) to (2, -3). <# **Goal**> Then $T$ maps the point (2, 1) to | <# **Context & Condition**> Let $T$ be the linear transformation from $R^2$ to $R^2$ such that $T$ maps (1, 2) to (2, 3) and (-1, 2) to (2, -3). <# **Goal**> Then, the linear transformation $T$ will map the point (2, 1) to |

## B.2   Format-level Perturbation: Question Format Refactoring

Examples of question format refactoring are shown in Table 8.

```
Here is a sentence in a multiple choice question.  Please rewrite the
sentence given its context and the expected similarity score.  Here
are necessary requirements:
[Requirements Start]
1.  Be consistent with its context.
2.  The rewrited sentence should keep the semantic of the original
sentence.
3.  If the sentence contains blanks/underlines to be filled, these
blanks/underlines should be kept after paraphrasing.
4.  You can utilize various rewriting skills (e.g.,
add/replace/delete words, paraphrase) to make it looks different from
the original.
[Requirements End]

[Meaning of Expected Similarity Score Start]
For the expected similarity score (0.0 - 1.0), 1.0 denotes that the
rewrited is exactly the same as the original; 0.8 denotes that the
the there exist word-level differences between the rewrited and the
original; 0.6 denotes that there exist not only word-level, but lots
of sentence structure-level differences between the rewrited and the
original; 0.4 denotes that you are allowed to entirely paraphrase
the sentence by your own; 0.2 denotes that you are allowed to add
misleading statements to the current sentence.
[Meaning of Expected Similarity Score End]

You should only output the rewrited sentence without any extra
content.
Expected similarity score:  {similarity_score}
Context:  {context}
Sentence:  {sentence}
Your output:
```

Figure 5: Prompt template for the rewriter LLM. The expected similarity score is used to control parphrasing of the rewriter LLM, which is set to 0.6 in all experiments.

Table 8: Examples of format-level knowledge-invariant perturbations. Texts surrounded by angular brackets are invisible in question prompts input to the LLM test-taker.

| Perturbation | Original case | Perturbed case |
|---|---|---|
| OptionPerm | <# **Options**> 
 A $x = 1$; B $x = 2$; C $x = 3$; D $x = 4$ | <# **Options**> 
 A $x = 4$; B $x = 3$; C $x = 2$; D $x = 1$ |
| OptionForm | <# **Options**> 
 A $x = 1$; B $x = 2$; C $x = 3$; D $x = 4$ | <# **Options**> 
 A) $x = 1$; B) $x = 2$; C) $x = 3$; D) $x = 4$ |
| OptionCaesar | <# **Options**> 
 A $x = 1$; B $x = 2$; C $x = 3$; D $x = 4$ | <# **Options**> 
 U $x = 1$; V $x = 2$; W $x = 3$; X $x = 4$ |
| ChangeType | <# **Prompt**> Please select correct option(s) given the following question: | <# **Prompt**> Please judge whether each of the options is correct given the following question: |
| SwapPos | <# **Prompt**> Please select correct option(s) given the following question: 
 <# **Question**> The solution of the equation $2x + 1 = 3$ is 
 <# **Options**> A $x = 1$; B $x = 2$; C $x = 3$; D $x = 4$ | <# **Prompt**> Please select correct option(s) given the following question: 
 <# **Options**> A $x = 1$; B $x = 2$; C $x = 3$; D $x = 4$ 
 <# **Question**> The solution of the equation $2x + 1 = 3$ is |

Table 9: The criteria of scores for knowledge invariance judgement.

| Score | Criteria |
|-------|----------|
| 5.0 | The perturbation perfectly satisfies all the standards and is entirely acceptable. |
| 4.0 | There are only minor flaws in the perturbed question. As a whole, the perturbed question is acceptable. |
| 3.0 | Only some parts of the perturbation is acceptable. As a whole, the perturbed question is less acceptable. |
| 2.0 | There are major flaws in the perturbed question that makes it unacceptable. |
| 1.0 | There are fatal flaws in the perturbed question that makes it entirely unacceptable. |

```
Your task is to grade the knowledge invariance degree of a perturbed
multiple choice question against the original question.
You clearly know that if a perturbed question is knowledge-invariant,
the question has to satisfy the following requirements:
[Perturbation Requirements Start]
1.  Semantic Information Invariance.  The perturbed question must
have the same semantic information as the original question, which
cannot change the name of entities, logic of statements and meaning
of equations.
2.  Reasoning Invariance.  A human test-taker's reasoning process
to obtain his/her response in the perturbed question should be
consistent with that in the original question.
3.  Answer Invariance.  The answer of a perturbed question should be
semantically equivalent to the answer of the original question.
4.  Statement Clarity.  The perturbed question should clearly present
contexts, conditions and the target of the question without ambiguous
statement.
[Perturbation Requirements End]

The grading score is from 1 to 5.  Grading criteria are given in the
following:
[Grading Criteria Start]
1.0 - There are fatal flaws in the perturbed question that makes it
entirely unacceptable.
2.0 - There are major flaws in the perturbed question that makes it
unacceptable.
3.0 - Only some parts of the perturbation is acceptable.  As a whole,
the perturbed question is less acceptable.
4.0 - There are only minor flaws in the perturbed question.  As a
whole, the perturbed question is acceptable.
5.0 - The perturbation perfectly satisfies all the requirements and
is entirely acceptable.
[Grading Criteria End]

[Original Question Start]:
{original_question}
[Original Question End]

[Perturbed Question Start]:
{perturbed_question}
[Perturbed Question End]
```

Figure 6: Prompt template for the LLM-based knowledge invariance scoring (Part 1/2).

```
You should grade the perturbation following these steps:
1.  Recall the perturbation requirements and grading criteria, and
read the original and the perturbed questions in detail.
2.  For each of perturbation requirements, carefully judge its
satisfaction degree of the perturbed question.
3.  Based on step 1 and step 2, give a total grading score for the
perturbed question.
4.  Analyze strengths and weakness of the perturbed question from the
view of perturbation requirements based on step 1,2,3.
Think carefully for a while, then propose your conclusion.  Your
output template is given as follows:

[Template Start]
{
"score":  <numeric score from 1 to 5>,
"strength":  <"xxx", strengths of the perturbation>,
"weakness":  <"xxx", weaknesses of the perturbation>
}
[Template End]

Your conclusion:
```

Figure 7: Prompt template for the LLM-based knowledge invariance scoring (Part 2/2).

### B.3 Correct Response Consistency

Similar to the confusion matrix, terms in ROP can be written in a response consistency matrix, as shown in Table 10. Here "$C$"/"$I$" in the first position denotes "consistently"/"inconsistently". "$C$"/"$W$" in the second position denotes "correct"/"wrong". For instance, $CC$ denotes *consistently correct*.

Table 10: Response transition matrix.

| Number | Correct after perturbation $\sigma$ | Wrong after perturbation $\sigma$ |
|---|---|---|
| **Correct before perturbation** $\sigma$ | $CC$ | $IC$ |
| **Wrong before perturbation** $\sigma$ | $IW$ | $CW$ |

## C  Experiment Setup

### C.1  Details of Dataset

Statistics of datasets are presented in Table 11.

Table 11: Dataset Statistics

| Name | College Mathematics | High School World History | Professional Psychology | Professional Medicine |
|---|---|---|---|---|
| Supercategory | STEM | Humanities | Social Science | Others |
| Concepts | Differential equations, real analysis, combinatorics... | Ottoman empire, economic imperialism, World War I... | Diagnosis, biology and behavior, lifespan development, ... | Diagnosis, pharmacotherapy, disease prevention... |
| # Questions | 100 | 237 | 612 | 272 |
| # Tokens per Q | 46.00±25.37 | 290.25±124.13 | 28.02±19.53 | 144.66±65.41 |
| # Tokens per P | 129.52±29.19 | 392.27±127.62 | 125.11±31.63 | 233.58±66.60 |

## C.2 Evaluation Setting

**Model Setting.** For close-sourced LLMs (gpt-4-turbo, gpt-3.5-turbo, glm-3-turbo, gemini-1.0-pro), we use their APIs for evaluation. The temperature parameter of each model is set to 0.2 to reduce output uncertainty. For open-sourced LLMs (llama-3-8b-instruct, mistral-7b-instruct-v0.2), we locally deploy their huggingface version to our own server and utilize default settings for evaluation.

**Environment Setting.** All experiments are completed on a Linux server with Intel(R) Xeon(R) Platinum 8269CY CPUs @ 2.50GHz and one NVIDIA A100 GPU (40G). GPUs are used for deploying and fine-tuning open-sourced models. The version of Python is 3.9. The version of the `torch` package is 2.2.0. The version of the `transformers` package is 4.39.3.

**Prompt Setting.** In our experiments, we use prompt templates to generate prompts of questions for evaluation. Examples using prompt templates are shown in Figure 8 and 9. After getting model outputs, we transform them to json data for further analyses and utilize regular expressions to enhance the robustness of transformation. Details are available at our code.

**Supervised Fine-tuning Setting.** In Appendix D.3, we fine-tune llama-3-8b-instruct using PertEval-generated data to explore the utility of PertEval in enhancing the knowledge capacity of LLMs. The fine-tuning process is done using LoRA [68] on single GPU implemented in `LLaMA-Factory` [69]. The batch size is 1, since the number of training data is 237 (High School World History) + 272 (Professional Medicine) = 479. The learning rate is 5e-5. The number of training epoch is 3.

```
Please select the correct option(s) from the following options given
the question:
Question:  Let k be the number of real solutions of the equation
e^x + x − 2 = 0 in the interval [0, 1], and let n be the number of real
solutions that are not in [0, 1].  Which of the following is true?
Options:
A k = 0 and n = 1
B k = 1 and n = 0
C k = n = 1
D k > 1
Your output must strictly follow this format:
{"answer":  <the list of selected options, e.g., ["A", "B", "C",
"D"]>}
Your output:
```

Figure 8: Prompt template for multiple-choice question answering.

```
Please judge whether each of the options is correct or incorrect
given the question:
Question:  Let k be the number of real solutions of the equation
e^x + x − 2 = 0 in the interval [0, 1], and let n be the number of real
solutions that are not in [0, 1].  Which of the following is true?
Options:
A k = 0 and n = 1
B k = 1 and n = 0
C k = n = 1
D k > 1
Your output must strictly follow this format:
{"A":  <"True" or "False">, "B":  <"True" or "False">, "C":  <"True"
or "False">, "D":  <"True" or "False">}
Your output:
```

Figure 9: Prompt template for multiple-judgement question answering (ChangeType).

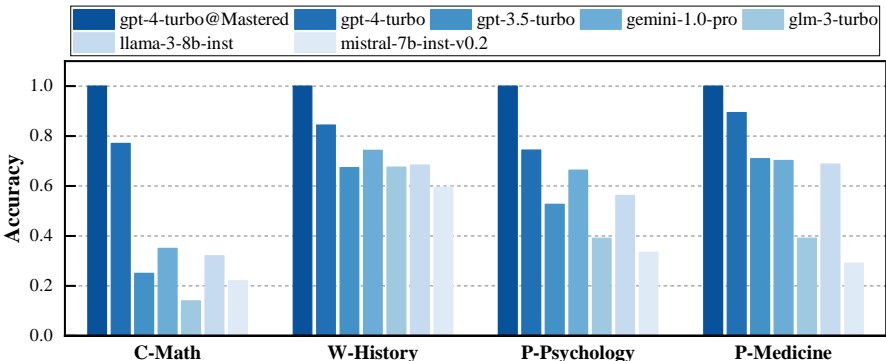

Figure 10: Accuracy score of LLMs in original datasets. The gpt-4-turbo@Mastered denotes the accuracy of gpt-4-turbo on the mastered question set for checking knowledge invariance, which is defined in Section 3.2.

## C.3 Experiment Cost

**Costs in # Tokens.** Overall, we conduct the following experiment steps that require token costs:

**E0.** Generating PertEval benchmark.

**E1.** Knowledge-invariance validation - obtain knowledge invariance score.

**E2.** Main experiment - obtain ACC@Consist.

**E3.** Fine-grained perturbation analysis - obtain ROP and PDR.

Exact costs w.r.t. experiments are presented as follows:

Table 12: Exact token costs w.r.t. experiments.

| Index | # Input tokens | # Output tokens |
|-------|----------------|-----------------|
| **E0** | 3,542,896 | 374,967 |
| **E1** | 776,353 | 101,675 |
| **E2** | 3,248,124 | 190,476 |
| **E3** | 9,343,200 | 336,996 |

**Costs in USD.** For proprietary LLMs, the cost is from the pricing strategy of LLM APIs (proportional to # input/output tokens). For open-sourced LLMs, the cost originates from the pricing strategy of servers for LLM deployment (proportional to the running time of servers). We count overall costs in USD per experiment, as shown in Table 13.

Table 13: Costs in USD per experiment.

| Exp Index | Price (USD) |
|-----------|-------------|
| **E0** | 41.79 |
| **E1** | 10.81 |
| **E2** | 13.09 |
| **E3** | 51.47 |
| **Total** | **117.16** |

## D   Verifying the Knowledge Invariance of Perturbation Strategies

In this part, we utilize the proposed knowledge invariance verification methods to measure the validity of perturbations in terms of knowledge invariance. The prompt template for the LLM-based

knowledge invariance scoring is presented in Figure 6 and 7. Results of **LLM-based knowledge invariance scoring** are shown in Figure 2. Results of **overall performance stability testing on mastered questions** are presented in Figure 4. We obtain several conclusions from these experimental results, as given in the following:

**C1. Proposed perturbations are knowledge-invariant in most cases.** From the right part of Figure 2, we observe that knowledge invariance scores of our proposed perturbations exceeds 4.5 and exceeds the baseline in most cases. For the content-level KnInvPara, since paraphrasing question confronts inevitable semantic changing, its knowledge invariance score is not as high as that of format-level perturbations. Indeed, we can observe from Table 4 that PDRs of KnInvPara are always closed to zero, which demonstrates its potential knowledge invariance.

**C2. Traditional string-based text distance metric is insufficient for measuring knowledge invariance.** In the left part of Figure 2, we show the average logarithm of Leveinshtein distance (edit distance) for each perturbation. Comparing the Levenshtein distances of PromptAttack, global text ordering-level format perturbation (ChangeType, SwapPos) and KnInvPara, we find that although PromptAttack has the lowest Leveinshtein distance in most datasets, its knowledge invariance scores are also relatively low. On the other hand, PromptAttack have the highest Leveinshtein distance in W-History, but also has the lowest knowledge invariance score in the four datasets. Therefore, As a result, there lacks an obvious correlation between Leveinshtein distance and knowledge invariance score. Essentially, this result indicates that knowledge relevant features of texts could be be disentangled from string-level features to some extent. How to efficiently and effectively measure knowledge invariance or quantify knowledge relevant features of texts remains a research problem.

**C3. LLMs are vulnerable to global text order-level knowledge-invariant perturbations even in mastered questions.** We observe from Table 4 that SwapPos can decrease the performance of gpt-4-turbo even in mastered questions although we have demonstrated the knowledge invariance of the perturbation in Figure 2, Comparing PDRs on different datasets and statistics of different datasets, we observe that it is highly correlated with the number of tokens per question.

## D.1 Probing LLMs' Consistent Knowledge Capacity

**Proposition D.1.** *In multiple choice questions, given $k$ options and one single correct answer for each question, the expected value of ACC@Consist is $1/k^2$ for pure guessing.*

*Proof.* Let $n$ be the number of multiple choice questions. To calculate ACC@Consist for pure guessing, we need to randomly select an option respectively for the original version and the perturbed version of each question. Since each selection procedure is mutually independent, the probability of selecting correct options for both the original and the perturbed versions is $1/k \times 1/k = 1/k^2$. Let random variable $Z$ denote the number of such cases, then $Z$ follows the binominal distribution $B(n, 1/k^2)$, and ACC@Consist $= Z/n$. Then the expected value of ACC@Consist is calculated by:

$$E[\text{ACC@Consist}] = E[Z/n] = \frac{1}{n}E[Z] = 1/k^2 \tag{4}$$

Then the proof is completed. $\square$

## D.2 Response Pattern Analysis for LLMs in PertEval

In this part, we analyze response pattern for LLMs in the original and perturbed data, which can provide us a deep insight into the influence of knowledge-invariant perturbations on LLMs' question answering procedure. The definition of response patterns are listed in the following:

- **Single Correct Selection**: The LLM selects and only selects the correct option. The ratio of correct choices is equvalent to micro accuracy.
- **Single Wrong Selection**: The LLM selects only a single option, and the option is wrong. This means that the LLM fails to recognize the correct option in the option list.
- **Invalid Selection**: The LLM selects none of options. This could be caused by the LLM's wrong reasoning process or invalid output format.
- **Multiple Selection Including the Correct One**: The LLM selects not only the correct option, but also extra incorrect options. This means that the LLM fails to filter out incorrect options from the option list.

- **Multiple Selection w/o the Correct One**: The LLM selects multiple options which do not cover the correct option.

Given the response pattern taxonomy, we counted the ratio of response patterns in original and perturbed data for all selected LLMs, as shown in Figure 4, 11, 12, 13, 14 and 15.

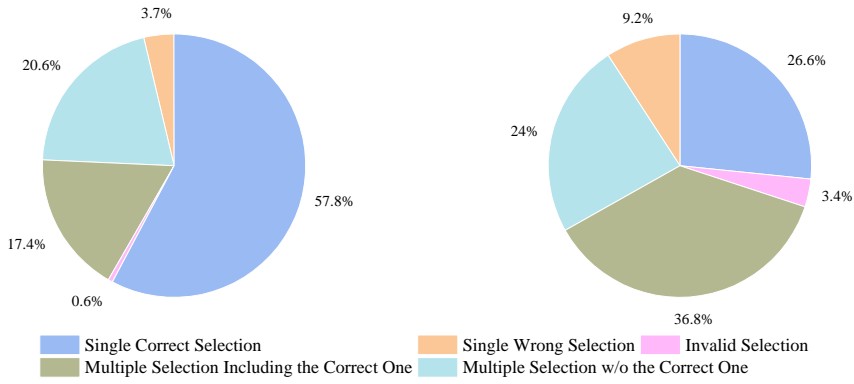

Figure 11: Response patterns of **gpt-3.5-turbo** in all datasets. Left: original data. Right: perturbed data.

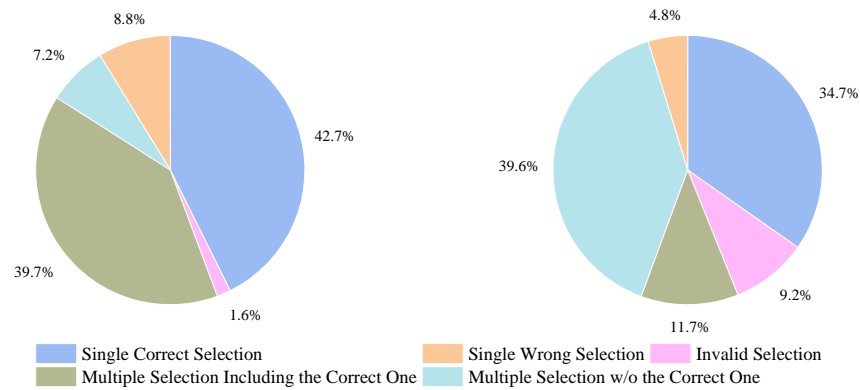

Figure 12: Response patterns of **glm-3-turbo** in all datasets. Left: original data. Right: perturbed data.

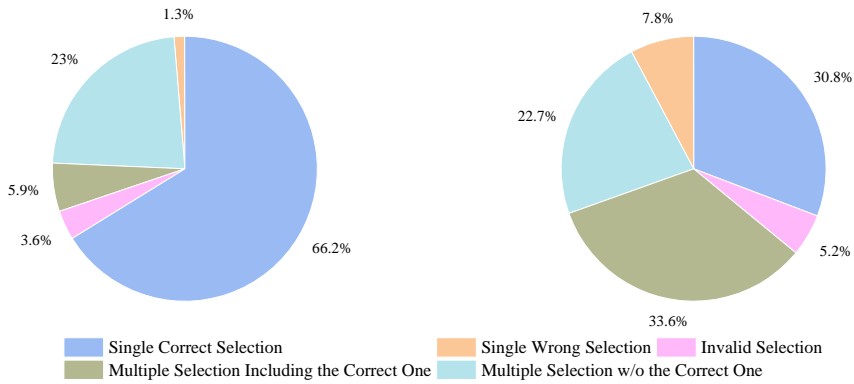

Figure 13: Response patterns of **gemini-1.0-pro** in all datasets. Left: original data. Right: perturbed data.

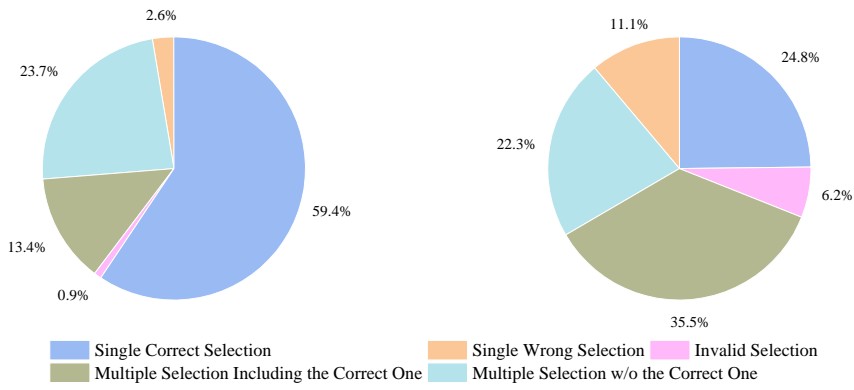

Figure 14: Response patterns of **llama-3-8b-instruct** in all datasets. Left: original data. Right: perturbed data.

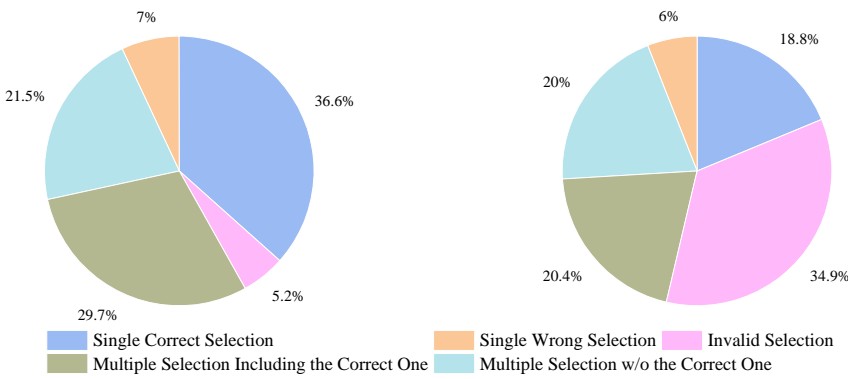

Figure 15: Response patterns of **mistral-7b-instruct-v0.2** in all datasets. Left: original data. Right: perturbed data.

### D.3   How to Enhance LLMs' Knowledge Capacity Using PertEval?

The goal of PertEval is not only to evaluate large language models' consistent knowledge capacity, but to help LLMs enhance their knowledge capacity in real-world applications. To this end, we explore utilizing perturbed data generated by PertEval to fine-tune the open-sourced LLM, LLaMA-8B-Instruct, as presented in Table 14. We obtain several insights from the experimental results.

For format-level perturbations, we recommend fine-tune LLMs on a small set of perturbed data to enhance their consistent knowledge capacity in all knowledge domains. This conclusion originates from the observed **stimulation phenomenon** in fine-tuning. That is, only fine-tuning the model with a subset of perturbed data can significantly improve its overall performance stability in all perturbed data. This result indicates the ability of LLMs to "acquire" format-level perturbations and perform consistently on perturbed data in all kinds of knowledge domains.

For content-level perturbations, we recommend training knowledge expert models using both the original and the perturbed data for specific knowledge domains. This conclusion orignates from **the lack of transferability of content-level fine-tuning** for datasets. This observation is reasonable because the knowledge and statement styles of different domain vary a lot. Considering the numerous number of knowledge domains and the issue of cost and efficiency, we suppose training knowledge expert models for specific knowledge domains is a good choice for striking a balance between knowledge capacity and training efficiency.

Table 14: PDR of LLaMA-8B-Instruct before and after fine-tuning on perturbed datasets. Underlined datasets are used for fine-tuning. CT denotes ChangeType. KP denotes KnInvPara. SP denotes SwapPos. For example, F(CT) means fine-tuning LLaMA-8B-Instruct on world history and professional medicine datasets perturbed by ChangeType.

| Strategy | fine-tune | C-Math | W-History | P-Psychology | P-Medicine | AVG$_{macro}$ | AVG$_{micro}$ |
|---|---|---|---|---|---|---|---|
| ChangeType | None | **-0.2300** | **-0.3418** | **-0.2467** | **-0.3493** | **-0.2920** | **-0.1998** |
| | F(CT) | -0.0700 | +0.0759 | +0.0196 | +0.0257 | +0.0128 | -0.0868 |
| | F(CT+KP) | -0.0500 | +0.0422 | +0.0082 | +0.0074 | +0.0020 | +0.0019 |
| SwapPos | None | **-0.0700** | **-0.2110** | **-0.1944** | **-0.2500** | **-0.1814** | **-0.2867** |
| | F(SP) | +0.0100 | -0.0675 | -0.1095 | -0.0882 | -0.0638 | +0.0246 |
| | F(SP+KP) | -0.0300 | -0.1350 | -0.1029 | -0.1176 | -0.0964 | -0.1065 |
| KnInvPara | None | +0.0200 | **-0.0802** | -0.0163 | **-0.0478** | -0.0311 | -0.0328 |
| | F(KP) | **-0.0400** | -0.0253 | -0.0212 | 0.0000 | -0.0216 | -0.0188 |
| | F(CT+KP) | **-0.0400** | -0.0549 | **-0.0343** | -0.0368 | **-0.0415** | **-0.0393** |
| | F(SP+KP) | -0.0300 | -0.0675 | -0.0212 | -0.0184 | -0.0343 | -0.0303 |

