# OpenReview forum: "PertEval: Unveiling Real Knowledge Capacity of LLMs with Knowledge-Invariant Perturbations"
_NeurIPS.cc/2024/Datasets_and_Benchmarks_Track — NeurIPS 2024 Track Datasets and Benchmarks Spotlight_

### Official Review · Reviewer_XdWc · 2024-07-17

**Rating:** 6
**Confidence:** 3

**Review:**

This paper introduces PerEval, an evaluation tool to provide an evaluation of LLM reliability and robustness with the concept of master questions for knowledge invariance. It is clearly written and seems original in the idea of master questions. However, one drawback of the evaluation is the lack of generalization testing on the generation of master questions.

**Strengths:**

1. The quality of this paper is good. It presents a detailed methodology for evaluating LLMs using PertEval to ensure reproducibility and transparency. The paper employs comprehensive validation techniques, including LLM-based knowledge invariance scoring and testing on mastered questions and it helps ensure the robustness of the evaluation.
2. The clarity of this paper is good. It is well-structured with clearly defined sections and makes it easy to follow the logical progression of the research. The paper also provides detailed explanations of the tables and figures.
3. The originality of this paper is good. It introduces PertEval and it’s a novel evaluation toolkit that uses innovative perturbation strategies and rigorous validation methods to provide a more systematic and comprehensive evaluation of LLM reliability and robustness.
4. The significance of this work is important. By addressing the limitations of traditional static benchmarks, PertEval provides a more accurate and reliable evaluation of LLMs' knowledge capacity. It may become a new guidance for evaluating the knowledge capacity of LLM.

**Additional Feedback:**

No.

**Clarity:**

The clarity of this paper is good. It is well-structured with clearly defined sections and makes it easy to follow the logical progression of the research. The paper also provides detailed explanations of the tables and figures.

**Correctness:**

The use of GPT-4 to generate master questions and evaluate with GPT-4 makes it less convincing in the evaluation.

**Documentation:**

Sufficient details are provided.

**Ethics:**

No.

**Limitations:**

Please see the improvement part.

**Opportunities For Improvement:**

1. It’s better to add up more figures and examples to help readers understand the key ideas.
2. Figure 1 (b) doesn’t have a detailed explanation.
3. The Master question is generated by gpt-4, so it might have a bias if you test knowledge invariance on gpt-4 turbo. It will be better to use other LLMs to generate master questions.

**Relation To Prior Work:**

Yes.

**Summary And Contributions:**

Regarding the concern of the limitation of test scenarios and the unavoidable risk of data contamination, this paper provides a new evaluation toolkit named PertEval to help evaluate the reliability of LLMs. PertEval offers innovative perturbation strategies and validation methods, allowing for a more systematic evaluation of the reliability of LLMs.

---

> ### Author Rebuttal · Authors · 2024-08-17
>
> We thank you very much for your constructive comments. We hope the following rebuttal could address your concerns.
>
> >Q1. Key points for help readers to better understand the key ideas.
>
> R1. Thank you very much for your thoughtful comment. Currently most examples are added in the appendix (see supplementary material) due to page limitation. We would add figures and examples to our code and the future version of the paper to make the work easier to understand.
>
> >Q2. Is there any detailed explanation of Figure 1(b)?
>
> R2. We described Figure 1(b) in line 239 – 241 in the paper. Let us explain in a more detailed manner. Levenshtein distance is a measure of the minimum number of single-character edits (insertions, deletions, or substitutions) required to transform one string into another. It is a classic character-level distance measurement of two texts. The motivation of comparing character-level distances and knowledge-invariance scores is to explore correlation between character-level and knowledge-level similarity. Results demonstrate that the knowledge-level similarity is largely different from character-level similarity, pointing the significance for exploring effective and efficient measurement of knowledge similarities of texts.
>
> >Q3. How to handle the potential bias in mastered questions generated by gpt-4 in the knowledge invariance verification experiment?
>
> R3. It should be clarified that mastered questions are **selected from the full dataset according to the performance of all LLMs** rather than gpt-4. Therefore the selection of mastered questions is fair. Plus, the rewriter LLM for KnInvPara is changable both in theory (our paper) and practice (our code). Therefore, we could generate various PertEval test data with different rewriter LLMs. The more excellent the rewriter LLM in natural language understanding, the higher the quality of the perturbed data.
>
> Please let us know whether our rebuttal address your concerns. Thank you very much for your time and patience.

---

> ### Author Response · Authors · 2024-08-30
> **Please Feel Free To Let Us Know If Any Concerns Remain Unresolved**
>
> Thank you for your thorough and insightful comments on our work. To address your concerns, we appended clarifications and experiments in our rebuttal. If you have any concerns that remain unresolved, please feel free to let us know. We will try our best to address your concerns. Thank you again for your time and patience!

---

### Official Review · Reviewer_GSDJ · 2024-07-21
**Interesting paper, missing literature review, positioning and evaluation**

**Rating:** 8
**Confidence:** 4

**Review:**

This paper addresses very important problems in robustness and contamination of LM evaluations with a promising method of knowledge invariant perturbation. However the key trait of knowledge invariance is validated only superficially. That said, the results in the paper do show a clear and alarming gap in state of the art model performance as revealed by their perturbations.

Pros:
- Clear performance gaps in strong models revealed by their perturbations
- A toolkit assembling LM-based paraphrasing and formatting perturbations

Cons:
- Claimed knowledge invariance undermined by very limited automatic validation, and no human evaluation.
- Lack of comparison and acknowledgement of previous work

**Strengths:**

1. A set of automatic perturbations of existing evaluation benchmarks.
2. The clear gap in performance between perturbed and unperturbed evaluations in figure 2 demonstrates their ToolKit’s ability to find lacking robustness even in the most advanced models.
3. Releasing code for their toolkit may enable quicker application of this technique to new models and benchmarks.

**Additional Feedback:**

- The introduction is very long, and somewhat repetitive. Consider shrinking it.
- The claim you evaluate on MMLU in the intro is misleading, as you only evaluate on a small subset of it.
- L.94 you mention GSM-8K as a knowledge evaluation, however this is not such a dataset as it evaluates math reasoning.
- Consider reporting time and costs for the proposed methodology
- The “Question type changing” paragraph (L.148) isn’t very clear
- L.198 you mention “complicated” - how is this measured?
- Table2: it’s hard to put these numbers in context. Can you run some baseline(s) to situate these numbers?
- Figure2: how much these results are specific to the chosen subsets of MMLU?

**Clarity:**

There are some key points that could greatly improve the writing.
- The paper lacks concrete examples, and leaves only a few to the appendix. For instance no examples of the kinds of perturbations they use are provided in the abstract or introduction, leaving the reader to discover only on the 3rd page that they use LM-based paraphrasing and format changes such as option permutation.
- The figure and tables are not self-contained by their captions and require extensive reference to terms defined elsewhere in the paper.
- There are bits of notation introduced and then not used again (line 166) that fall short of providing a full formal definition of a method. Terms such as “option ID” (line 136) are used without definition or example.
- Also Important information is frequently left to the appendix leaving important methods only vaguely defined in the body of the paper: For example, the knowledge-invariant paraphrasing described on line 119 does not include key details such as what specific LM is used. Also the body of the paper doesn’t state what inference format is assumed for the unperturbed evaluation (e.g., are answers ranked based on likelihood, or are letter-mapped answers generated and matched).

**Correctness:**

The authors claim to provide “knowledge-invariant perturbations” which “meticulously [retain] knowledge-critical content” (line 6-8). They do provide tentative evidence in support of this claim with the two previously discussed validation methods. But without manual analysis and discussion of limitations of these validation methods, I am not convinced that this very strong claim can be made. That said, I think simply reducing this claim to a softer “attempt” to create relatively “more” knowledge invariant perturbations would more accurately describe the good contributions that are still in the paper.

**Documentation:**

The authors contribute an evaluation toolkit. Their URL is available and leads to a repo with a readme and code. The readme provides a few sentences describing the various files. The repo does not however provide specification of the python packages used or any instruction for setting up your environment.

**Limitations:**

The authors address limitations such as the limited number of benchmarks, and non-applicability of their approach to open-ended benchmarks. However, a discussion of the limitations to their ability to validate the knowledge invariance of their perturbations should also be included. Validating knowledge invariance is a very challenging task and even if it is not possible to guarantee this, the paper explores some interesting ways to gain some assurance about knowledge invariance. However without more discussion of the limitations, the paper risks prematurely claiming to have solved this very difficult problem.

**Opportunities For Improvement:**

The validation of knowledge invariance relies on two adhoc automatic tests and no manual analysis is provided in the body of the paper. While exploring automatic LM-based validation of knowledge invariance is an interesting approach, this validation approach can’t just be assumed to work at face value without being validated itself. The alternative validation approach is ensuring non-degradation of examples that are sufficiently easy that no models get them wrong before perturbations. This approach is interesting, but at most only addresses what the authors describe as “answer invariance” without considering if the underlying knowledge required is or is not changed. It also doesn’t test if perturbations have accidentally made the questions easier (for example by including the answer in the question).

There’s often a lack of clarity regarding the details and formula used in the paper. It would be super helpful to describe in text the intuition behind each formula and what it is intended at capturing. Same comment holds on the different perturbations as well. In a similar vein, it would be helpful to provide concrete examples in the main paper, rather than the appendix.

The paper over claims regarding its evaluation on MMLU, as it only evaluates a small portion of this dataset. The smaller subset use is not a limitation on its own, but this should be discussed much earlier in the paper. Similarly, it isn’t clear how transferable the perturbations are on other subsets of MMLU, and other datasets in general.
Ideally, I’d propose the authors to use a small set of examples from each MMLU subset, rather than all examples from specific subsets, as this would be more representative on the MMLU results, as proposed in https://arxiv.org/abs/2402.14992. However, if this is out of budget, it would also be acceptable to just rewrite the mentions of evaluations on MMLU as clearly being “4 subsets of MMLU.”

**Relation To Prior Work:**

The authors provide a related works section with extensive citations. However it remains unclear to me how exactly they situate their work. They state on line 105, “Differently, PertEval aims to improve trustworthiness in capacity assessment through extensively evaluating LLMs on existing benchmarks with perturbations.” It is unclear if this statement distinguishes them from all work they have just mentioned in “trustworthiness of knowledge capacity evaluation” or just the last one. Probably most importantly they claim their method “is built upon a new concept of knowledge-invariant perturbation” (line 113). This is incorrect.
The paper misses papers from the NLP literature that should be further researched, cited and compared to.
- https://arxiv.org/pdf/2005.04118
- https://arxiv.org/pdf/2101.00288

Another important body of work that is not referred to is consistency of language models, that uses similar metrics, and ideas to evaluate the knowledge/robustness of models to knowledge/capabilities.
- https://arxiv.org/abs/2102.01017
- https://aclanthology.org/2022.coling-1.324.pdf
- https://aclanthology.org/2022.findings-acl.240/
- https://arxiv.org/abs/2305.14279




In terms of quantitative comparison to previous work or ablations of their own method design, they only compare to one baseline perturbation method, PromptAttack, and then only in their knowledge invariance validation. It is understandable that further experiments may be too expensive; an alternative to this would be to better clarify which of the format perturbations have been tried in previous work or not and how those findings aligned with the findings in this paper.

**Summary And Contributions:**

This paper designs, validates, and performs analysis with a toolkit of knowledge invariant perturbations for closed-end benchmarks. Examining differences in model responses due to these perturbations can reveal lacking model generalization and reliance on memorization or even benchmark contamination. Their perturbations include sentence-by-sentence paraphrasing using a prompted large LM and format perturbations such as permuting the options. They propose two approaches for validating the knowledge invariance of their permutations: 1) using a large LM to score knowledge invariance based on a rubric, 2) ensuring similar performance before and after perturbation of all test questions simple enough to be answered correctly by all examined models before perturbation. Based on good scores from these two validation methods they claim knowledge invariance for their perturbations. Finally, they use their perturbations to examine 6 contemporary models using 4 subsets from MMLU. They find clear inconsistencies between original and perturbed answers when composing their perturbations, and also examine the impacts of type of perturbation individually, finding many to be statistically significant.


---

Based on the author's response, I increased my score from 6 to 8.
This is based on the assumption that the authors will properly address the concerns I raised, further provide additional details (like they have in the response), and provide a more detailed and elaborate on related work in this topic.

---

> ### Author Rebuttal · Authors · 2024-08-17
>
> We thank you very much for your very constructive and insightful comment. We hope the following rebuttal could address your concerns.
>
> >Q1. Could the automatic knowledge invariance validation be more comprehensive?
>
> R1. Yes. It should be clarified that we managed to ensure the knowledge invariance of PertEval by not only ad-hoc validation methods, but also pre-hoc design. The example of including answer in the question is prevented based on our design. The answer is invisible for the rewriter LLM, which could effectively avoid including answers. Human evaluation is a direct and effective way for validating knowledge-invariance. Difficulties in human evaluation mainly lies in the mastery of professional knowledge and the consistency of scoring. To address your concern, we append human and other LLMs’ knowledge-invariance validation experiments. See “Common Rebuttal”.
>
>
> >Q2. What distinguishes this work from previous work, especially compared to the NLP literature?
>
> R2. We suppose the concept of "knowledge-invariant perturbation" is what distinguishes this work from previous work. Previous works (e.g., https://arxiv.org/pdf/2005.04118) indeed proposed perturbation-based method for evaluating specific abilities (e.g., counterfactual reasoning) of language models. However, compared to previous work, this work propose for the first time **the idea of "knowledge invariance" for knowledge capacity evaluation**. Moreover, PertEval is more general and flexible because it could be applied to various close-ended knowledge evaluation benchmarks, without rigorous requirement for text format (e.g., the text must be narrative).
>
> >Q3. How is the time and costs for the proposed methodology?
>
> R3. We would like to report costs because there exists certain procedures to obtain costs. Costs in PertEval mainly lies in generating perturbed data and the evaluation on the original and perturbed datasets. Overall, we have 1,221 test questions. Each question have eight versions, including one original, six obtained via atomic perturbation, and one obtained via composite perturbation. Each version is answered by every LLM. Here we show its approximate token costs:
>
> Table 1. Approximate Costs for knowledge-invariant paraphrasing (KnInvPara)
> |Name | Value |
> |:---|:---
> |# Input tokens | 3,500,000
> |# Output tokens | 370,000
>
> Table 2. Approximate costs for input tokens in obtaining results in Figure 2 in the paper. The number of tokens is calculated via OpenAI's toknizer, the TikToken.
>
> |Name|Value|
> |:---|:---
> |# Input tokens @ Original| 250,000
> |# Input tokens @ Atomic Perturbations | 1,530,000
> |# Input tokens @ PertEval-All| 290,000
> |# Input tokens for one LLM | 2,070,000
> |**# Input tokens for all LLMs** | **12,420,000**
>
> Overall, the number for input tokens in experiments is approximately 3,500,000 + 12,420,000 = 15,920,000. The number of output tokens is about 370,000, mainly from knowledge-invariant paraphrasing.
>
> >Q4. L.198 you mention “complicated” - how is this measured?
>
> R4. One measurement is the leveinshtein distance (the edit distance), which represents charachter-level difference between the original and the perturbed data. We present the edit distance for each atomic perturbation in Figure 1 (right), p6 in our paper.
>
> >Q5. How to put numbers in Table 2 in context? Can you run some baseline(s) to situate these numbers?
>
> R5. Yes. The best way to situate these numbers is to compare them with the ideal PDR value, zero, as mentioned in line 242 -- 243 in our paper. In addition, it would be rational if a baseline sometimes achieve good PDR because baselines usually apply small perturbations (e.g., word level) to texts. To better address your concern, we take PromptAttack as a baseline to compare with these results:
>
> Table 3. Performance Drop Rate (PDR) of GPT-4-Turbo on mastered questions with the PromptAttack perturbation.
>
> |Metric|C-Math|W-History|P-Psychology|P-Medicine|AVG$_{macro}$
> |:---|:---|:---|:---|:---|:---
> |Performance Drop Rate|0.0000|-0.2874|-0.0364|0.0000|-0.0810
>
> >Q6. How much results of Figure 2 are specific to the chosen subsets of MMLU?
>
> R6. Evaluation results on original datasets w.r.t. subjects are available at appendix Figure 9. ACC@Consist w.r.t subjects are presented as follows:
>
> Table 4. ACC@Consist w.r.t. subjects and LLMs.
>
> | **LLM**         | **C-Math** | **W-History** | **P-Psychology** | **P-Medicine** |
> |-----------------|------------|---------------|------------------|----------------|
> | gpt-4-turbo     | 0.4700     | 0.5570        | 0.5098           | 0.6838         |
> | gpt-3.5-turbo   | 0.0600     | 0.2447        | 0.1944           | 0.2537         |
> | glm-3-turbo     | 0.0500     | 0.3249        | 0.1977           | 0.1250         |
> | gemini-1.0-pro  | 0.0900     | 0.2911        | 0.2500           | 0.2757         |
> | llama-3-8b      | 0.0300           | 0.2236              | 0.1732                 | 0.2463              |
> | mistral-7b-v0.2 | 0.0100       |  0.1730             |  0.0915            |  0.0662              |
>
> Most LLMs except gpt-4-turbo perform very poor on C-Math. This phenomenon was also observed in the original dataset (see Figure 9, p22 in Appendix), where the accuracy of most LLMs on C-Math is close to 0.25, the random guessing score. Therefore, doubts could raises that whether these LLMs indeed hold math knowledge capacity.
>
> > Q7. Key points for improving writing.
>
> R7. We thank you very much for your very constructive and thoughtful comments. We believe that your comments could greatly improve the writing, especially making the study easier for readers to undersand. Therefore, we would follow your comments to clarify the statment about MMLU in introduction and put more key information in the main paper in the future. Following your constructive comment, we have also provided the guidance for setting up running environments in our code.
>
> Please let us know if our rebuttal address your concerns. Thank you very much for your time and patience.

---

> > ### Comment · Reviewer_GSDJ · 2024-08-22
> >
> > Thank you for addressing my review.
> >
> > > Human eval
> >
> > This is a great analysis! While the presentation wasn’t clear to me and should be improved for the camera-ready version, I think it’s a nice addition.
> > In addition, I think the main question is what happens to human performance when answering the modified benchmark to get an upper bound on performance. I understand collecting is more expensive, but that would be the optimal evaluation.
> >
> > > Relation to previous work
> >
> > While the papers you discuss in the response don’t look at knowledge invariance, the other papers I mentioned do. It is important to put this work in context with previous literature. I would emphasize again the importance of doing a proper literature review and discussing the similarities and differences between your and previous work.
> >
> > > Time and costs
> >
> > Thank you for providing this information. For completeness, could you also add the exact costs per experiment?
> >
> > > Other comments/suggestions
> >
> > Thank you for providing these details. They are helpful and would strengthen the paper when included in the next version.

---

> > > ### Author Response · Authors · 2024-08-26
> > > **Response to Reviewer**
> > >
> > > We thank you very much for your positive feedback.
> > >
> > > > Comments about human eval
> > >
> > > Thank you for your approval of our human-based knowledge invariance evaluation. We will improve the presentation of this experiment in the camera-ready version. For human performance comparison, the ideal experimental setting requires two classes of students with similar knowledge capacities, each taking an exam with either the original/modified benchmark. Therefore, we could consider cooperating with high schools and universities to conduct these experiments in the future.
> > >
> > > > Comments about relations to previous work
> > >
> > > We thank you very much for your constructive feedback. We will improve the literature review following your comments. We will discuss in detail the similarity and difference between ours and previous works in terms of research goal, methodology and results.
> > >
> > > > For completeness, could you also add the exact costs per experiment?
> > >
> > > Yes. Overall, we conduct the following experiments that require token costs:
> > >
> > > E0. Generating PertEval benchmark.
> > >
> > > E1. Knowledge-invariance validation - obtain KIScore.
> > >
> > > E2. Main experiment - obtain ACC@Consist.
> > >
> > > E3. Fine-grained perturbation analysis - obtain ROP and PDR.
> > >
> > > Exact costs w.r.t. experiments are presented as follows:
> > >
> > > Table 1. Exact token costs w.r.t. experiments.
> > > |Exp Index|# Input token|# Output token
> > > |:---|:---|:---
> > > |E0|3,542,896|374,967
> > > |E1|776,353| 101,675
> > > |E2|3,248,124|190,476
> > > |E3|9,343,200|336,996
> > >
> > > We thank you again for your constructive and thoughtful comments. We will add these details in the next version of our paper. Please let us know whether our response address your concerns.

---

> > > > ### Comment · Reviewer_GSDJ · 2024-08-26
> > > >
> > > > Thank you.
> > > > I increased my score from 6 to 8.
> > > >
> > > > Please make sure to address these comment in the next version. I'm excited to read the final version in a few months!
> > > >
> > > > p.s. the cost experiment you provided in Table 1 still does not provide the actual cost. Please indicate the cost in $ per experiment.

---

> > > > > ### Author Response · Authors · 2024-08-27
> > > > > **Response to Reviewer**
> > > > >
> > > > > We thank you very much for your positive feedback! We would add the improved literature review and experiment results to the next version. Meanwhile, we would improve the presentation and clarify confusing statements.
> > > > >
> > > > > > How is the cost in $ per experiment?
> > > > >
> > > > > Sorry for the confusion. For proprietary LLMs, the cost is from the pricing strategy of LLM APIs (proportional to # input/output tokens). For open-sourced LLMs, the cost is based on the pricing strategy of servers for LLM deployment (proportional to server running time). We count overall costs in USD per experiment, as shown in Table 1. The result will be added to the next version of the paper.
> > > > >
> > > > > Table 1. Costs in USD per experiment
> > > > > |Exp Index|Price (USD)|
> > > > > |:---|:---
> > > > > |E0|41.79
> > > > > |E1|10.81
> > > > > |E2|13.09
> > > > > |E3|51.47
> > > > > |__Total__|__117.16__
> > > > >
> > > > > We thank you again for your very constructive, detailed and thoughtful comments. We would follow your comments seriously to improve the paper writing and experiment analysis. We hope this study can provide insights for LLM researchers and boost the development of model evaluation and trustworthy ML in the future.

---

### Official Review · Reviewer_SFLD · 2024-07-22
**Interesting paper, but the presentation and the evaluation need to be improved**

**Rating:** 6
**Confidence:** 3
**Clarity:** Yes. The logic is coherent, and the s…

**Review:**

**Pros**
- The paper studies an interesting problem. Understanding whether the LLMs have actually acquired knowledge is important for trustworthy AI.
- The paper designs  a toolkit called PertEval that can probe the in-depth knowledge of LLMs.
- The paper's findings are interesting.

**Cons**
- The evaluation is based on GPT-4 which can be biased and not consistent with human evaluation. Therefore human evaluation may be acquired.
- The authors should discuss the consequences and solutions if, after publication, the LLMs are adaptively optimized against PertEval.

**Strengths:**

This work takes into account two shortcomings in the existing evaluation benchmarks for the knowledge capabilities of large language models (LLMs) and proposes a feasible and practical solution. The work is very interesting and has considerable potential for future development, making it worthy of further in-depth research in the relevant field. The paper is logically clear, and the expression is precise.

**Additional Feedback:**

Please refer to my comments above.

**Correctness:**

Basically the claims made in this paper are correct and reasonable. Some questions regarding the experimental setup have been specifically elaborated in the“Opportunities For Improvement” section above.

**Documentation:**

Yes there is sufficient detail for the benchmark.

**Limitations:**

Yes, the authors have adequately addressed the limitations and potential negative societal impact.

**Opportunities For Improvement:**

The paper may require human evaluation to justfy the GPT-4 evaluation. To test whether LLMs are merely memorizing or actually understanding the meaning, the work proposes a method of rephrasing questions while maintaining their actual meaning.
In Section 3.2, to verify whether the rephrased questions hold the same meaning as the original ones, GPT-4 Turbo is used as the judge. I have the following concerns: Does this verification require the assistance of human candidates? Otherwise, the scores for other LLMs' responses to this set of questions will be relative to GPT-4 Turbo's scores, which may not be very reliable. The limitations of this question set are worth considering. This experimental setup raises the suspicion that the examiner and the examinee are the same entity.

Additionally, the evaluation only includes small-sized LLMs. Additional evaluation on larger LLMs and MoE models is needed.

**Relation To Prior Work:**

Yes the paper has clearly discussed the difference from previous contributions.

**Summary And Contributions:**

This article proposes a toolkit named PertEval, which evaluates the knowledge capability of LLMs based on knowledge-invariant perturbations, serving as an extension of traditional close-ended evaluation benchmarks. This toolkit is then used to re-evaluate six representative LLMs.

---

> ### Author Rebuttal · Authors · 2024-08-17
>
> We thank you very much for your constructive and thoughtful comment. We hope the following rebuttal could address your concerns.
>
> >Q1. Does the verification of knowledge invariance require the assistance of human candidates?
>
> R1. The assistance of human candidates is optional in this verification. Indeed human evaluation is a general and direct verification method. Challenges in human-based knowledge invariance scoring mainly lies in the requirement of mastered professional knowledge and the consistency of human scoring.
>
> To further address your concern, we conduct experiements to compare knowledge invariance scores rated by humans different superior LLMs. See “Common Rebuttal”.
>
> >Q2. What are the consequences and solutions if, after publication, the LLMs are adaptively optimized against PertEval?
>
> R2. The risk of adaptive optimization against PertEval indeed exists because it is an open-sourced toolkit. However, the diversity of perturbations and the cost of adaptive optimization could effectively reduce the influence of such procedure. For instance, as introduced in line 131 – 158 in our paper, perturbations implementations could be implemented in various versions (e.g., A/B/C/D->U/V/W/X or A/B/C/D -> $\beta/\gamma/\sigma/\eta$). This mechanism has already been implemented in our code. Besides, results of knowledge-invariant paraphrasing are variant and invisible until the evaluation process starts because the perturbation is instantly generated by the rewriter LLM. Moreover, atomic perturbations could be combined together to obtain various composite perturbations. It is demanding for adaptive optimization to cover all possible combination of perturbations.
>
> Second, we could introduce new perturbations (that could pass the knowledge-invariance validation) based on the PertEval protocol. In term of both theory (in our paper) and practice (in our code), current perturbations are only implementations of the PertEval protocol. Developers are encouraged to introduce their own content-level or format-level strategies.
>
> Last but not least, we hope PertEval could be used not only for trustworthy LLM evaluation, but also for improving LLMs' knowledge capacity. Therefore, developers could use PertEval to fine tuning LLMs to enhance their robustness against various conditions. On another hand, the evaluation environment could be generated on-the-fly by PertEval, which could be different from the fine-tuning environment.
>
> > Q3. Additional evaluation on larger LLMs and MoE models is needed.
>
> R3. Thank you for your thoughtful comment. In our experiment, proprietary LLMs such as gpt-4-turbo and gpt-3.5-turbo are larger and MoE models. For open-sourced models, we choose llama-3-8b-instruct and mistral-7b-v0.2 considering the local computational cost for larger open-sourced LLMs. We will build an open evaluation platform based on the open-sourced PertEval that enables LLM developers from all around the world to test the knowledge performance of various LLMs online.
>
> We would add the discussion to our paper. Please let us know whether our rebuttal address your concerns. Thank you very much for your time and patience.

---

> ### Author Response · Authors · 2024-08-30
> **Please Feel Free To Let Us Know If Any Concerns Remain Unresolved**
>
> Thank you for your thorough and insightful comments on our work. To address your concerns, we appended clarifications and experiments in our rebuttal. If you have any concerns that remain unresolved, please feel free to let us know. We will try our best to address your concerns. Thank you again for your time and patience!

---

### Author Rebuttal · Authors · 2024-08-17

We appreciate all the reviewers' thorough assessment and valuable feedbacks. Their thoughtful evaluation has provided valuable insights that have significantly contributed to the improvement of the manuscript. The reviewers' positive comments encompassing different dimensions are truly encouraging:

- **Contribution**: "The work is very interesting and has considerable potential for future development, making it worthy of further in-depth research in the relevant field." (SFLD); "This paper addresses very important problems in robustness and contamination of LM evaluations with a promising method of knowledge invariant perturbation." (GSDJ); "It may become a new guidance for evaluating the knowledge capacity of LLM." (XdWc)
- **Method**: "The paper designs a toolkit called PertEval that can probe the in-depth knowledge of LLMs." (SFLD); "Releasing code for their toolkit may enable quicker application of this technique to new models and benchmarks." (GSDJ); "It introduces PertEval and it’s a novel evaluation toolkit that uses innovative perturbation strategies and rigorous validation methods to provide a more systematic and comprehensive evaluation of LLM reliability and robustness." (XdWc)
- **Experiment**: "The paper's findings are interesting." (SFLD); "The clear gap in performance between perturbed and unperturbed evaluations in figure 2 demonstrates their ToolKit’s ability to find lacking robustness even in the most advanced models." (GSDJ); "The paper employs comprehensive validation techniques, including LLM-based knowledge invariance scoring and testing on mastered questions and it helps ensure the robustness of the evaluation." (XdWc)
- **Presentation**: "The logic is coherent, and the sentences are clear." (SFLD); "It is well-structured with clearly defined sections and makes it easy to follow the logical progression of the research." (XdWc); "The paper also provides detailed explanations of the tables and figures." (XdWc)

Regarding the questions raised by each reviewer, we have carefully considered each point and have made detailed responses in the local rebuttal pane.

---

**[More detailed experiment of knowledge invariance validation]**

**Experiment Setup.** We select questions with id = 0, 5, 10, 15, 20, 25, 30, 35, 40, 45 for each subject (4 * 10 * 2 = 80 scoring sample pairs in total) as the scoring data. We composite all the six atomic perturbations together to obtain the PertEval-All. We recruit volunteers that master the required knowledge and train them face-to-face for human evaluation. We randomly divide them into three groups. Within each group, each volunteer rates for 40 randomly assigned samples.

Table 1. **Knowledge invariance scores$\uparrow$ (1-5) rated by human** (4 independent scores presented in ascending order).

| **Perturbation** | **C-Math**     | **W-History**  | **P-Psychology** | **P-Medicine** |
|------------------|----------------|----------------|------------------|----------------|
| PromptAttack$\uparrow$     | 2.3/2.4/3.0/3.9| 2.2/2.2/2.3/2.8 | 1.3/2.4/2.8/2.8 | 1.6/2.5/3.5/3.6 |
| **PertEval-All (ours)**$\uparrow$   | **3.6/3.8/3.9/3.9** | **3.7/4.1/4.1/4.3** | **4.3/4.4/4.5/4.7**   | **4.2/4.3/4.4/4.6** |

Table 2. **Knowledge invariance scores$\uparrow$ (1-5) rated by LLMs**. Values (a/b/c) in each cell denotes the average knowledge invariance score rated by GPT-4-Turbo, Claude-3.5-Sonnet, and LlaMA-3.1-405b, respectively.

| **Perturbation** | **C-Math**     | **W-History**  | **P-Psychology** | **P-Medicine** |
|------------------|----------------|----------------|------------------|----------------|
| PromptAttack$\uparrow$      | 3.2/3.6/3.6 | 3.2/3.3/3.7 | 3.9/3.9/3.7   | 4.1/4.3/4.2 |
| **PertEval-All (ours)**$\uparrow$    | **3.8/3.9/4.0** | **4.0/4.2/4.0** | **4.0/4.4/4.0**  | **4.1/4.4/4.0** |

Table 3. **Edit distance$\uparrow$** between the perturbed and the original questions.

| **Perturbation** | **C-Math**     | **W-History**  | **P-Psychology** | **P-Medicine** |
|------------------|----------------|----------------|------------------|----------------|
| PromptAttack$\uparrow$    | 34 | 642 | 72 | 3 |
| **PertEval-All (ours)**$\uparrow$   | **428** | **1090** | **379**   | **980** |

Table 4. **Number of human-imperceptible perturbations$\downarrow$**. The format is A/B. Here A denotes the number of human-imperceptible perturbed questions, i.e., no perturbation / only change line breaks and spaces. B denotes the number of all tested samples. These samples are removed from the calculation of KI score.

| **Perturbation** | **C-Math**     | **W-History**  | **P-Psychology** | **P-Medicine** |
|------------------|----------------|----------------|------------------|----------------|
| PromptAttack$\downarrow$      | 2/10 | 1/10 | 2/10   | 2/10 |
| **PertEval-All (ours)**$\downarrow$    | **0/10** | **0/10** | **0/10**  | **0/10**

**Analysis.** According to Table 1 and 2, PertEval-All outperforms the baseline, and the score mostly exceeds 4.0, the borderline of knowledge-invariance. Scores of both PromptAttack and PertEval-All on C-Math do not exceed 4.0 on C-Math. This is because many data in college_mathematics_test of MMLU are short mathematical reasoning questions that have many mathematical symbols and statements. In many STEM subjects, requirements for knowledge capacity and reasoning ability often mix together in a test question. This points out considerable potential for future development in robust LLM evaluation in STEM subjects.

Jointly analyzing Table 3 with Table 1 and 2, we find that PertEval could maintain high knowledge invariance scores even though the edit distance is very large ($\geq 379$). This reflects the essential difference between "string distance" and "knowledge distance". Indeed, how to effectively and efficiently measure knowledge distance between questions for LLM evaluation is a valuable future research problem.

---

### Decision · Program_Chairs · 2024-09-26

**Decision:**

Accept (Spotlight)

**Comment:**

This paper proposed,  PertEval, to evacuate the real knowledge capacity of LLMs. All reviewers provide positive scores for this paper and acknowledge the importance of the benchmark and novelty of the methods. AC read all rebuttals and reviewers comments. AC agrees with them.  AC hopes the authors can address the concerns in the final version